# Single cell sequencing reveals that CD39 inhibition mediates changes to the tumor microenvironment

Lilong Liu[1,4], Yaxin Hou[2,4], Changqi Deng[1], Zhen Tao [3] ✉, Zhaohui Chen [1] ✉, Junyi Hu[2] ✉ & Ke Chen [2] ✉

Single-cell sequencing technologies have noteworthily improved our understanding of the genetic map and molecular characteristics of bladder cancer (BC). Here we identify CD39 as a potential therapeutic target for BC via single-cell transcriptome analysis. In a subcutaneous tumor model and orthotopic bladder cancer model, inhibition of CD39 (CD39i) by sodium polyoxotungstate is able to limit the growth of BC and improve the overall survival of tumor-bearing mice. Via single cell RNA sequencing, we find that CD39i increase the intratumor NK cells, conventional type 1 dendritic cells (cDC1) and CD8 + T cells and decrease the Treg abundance. The antitumor effect and reprogramming of the tumor microenvironment are blockaded in both the NK cells depletion model and the cDC1-deficient *Batf3*[−/−] model. In addition, a significant synergistic effect is observed between CD39i and cisplatin, but the CD39i + anti-PD-L1 (or anti-PD1) strategy does not show any synergistic effects in the BC model. Our results confirm that CD39 is a potential target for the immune therapy of BC.

Urinary bladder cancer (BC) is one of the most prevalent cancers worldwide, with an estimated 81,400 new cases diagnosed and 17,980 deaths caused in the United States in 2020[1]. Approximately 75% of newly diagnosed BCs are nonmuscle-invasive (NMIBC)[2], while the remaining 25% were diagnosed at latter stages with muscle-invasive (MIBC) or metastatic disease[3]. NMIBC is generally not life-threatening, but the recurrence rate is as high as 50–70%[4]. Although the treatment and early diagnosis have substantially improved, BC-related mortality is still high[5]. The tumor-specific survival rate of high-grade NMIBC patients is ~70–85% in 10 years, and ~6% of patients with low grade and 17% of patients with high-grade NMIBC will progress to MIBC in the long-term follow-up[6]. However, MIBC is characterized by rapid progression, easy metastasis, poor prognosis and low survival rate[7], and the tumor-specific survival rate of MIBC patients is ~49–52% in 5 years[8].Thus, further

improved diagnostic and therapeutic techniques for BC patients are clearly needed.

For NMIBC, local tumor resection followed by intravesical bacillus Calmette-Guérin (BCG) immunotherapy has been effectively used in the last 30 years[9]. However, even after BCG treatment, with a median follow-up of 5.2 years (interquartile range: 2.6–8.5), 51% of the patients had a recurrent disease and 19% progressed to muscle-invasive disease[10].In contrast, the systemic management of muscle-invasive and advanced bladder cancer involves extirpation of the entire organ (cystectomy) and urinary diversion based primarily on platinum chemotherapy[11,12]. However, not all patients respond to cisplatin (CIS)-based neoadjuvant chemotherapy, and only 50% of patients could benefit from conventional chemotherapy[11]. There is an increasing interest in immune checkpoint blockade (ICB) therapy, especially in the USA and most of Europe, as the anti-PD1 (aPD1) and anti-PD-L1

---

[1]Department of Urology, Union Hospital, Tongji Medical College, Huazhong University of Science and Technology, Wuhan, China. [2]Department of Urology, Tongji Hospital, Tongji Medical College, Huazhong University of Science and Technology, Wuhan, China. [3]Department of Radiation Oncology and Cyberknife Center, Tianjin Medical University Cancer institute & Hospital, Tianjin, China. [4]These authors contributed equally: Lilong Liu, Yaxin Hou.
✉e-mail: ztao@tmu.edu.cn; zhaohuichen@hust.edu.cn; 531572782@qq.com; shenke@hust.edu.cn

(aPD-L1) treatments have gradually become part of the standard for patients with locally advanced or metastatic urothelial cancer who relapse after CIS-based chemotherapy or are considered CIS ineligible, but the objective response rate is only 20%[13]. Recently, we have generated a single-cell atlas of 8 tumor samples and 3 paratumor samples from BC patients and revealed distinct components and microenvironment features between different molecular subtypes, which may lead to differential responses to chemotherapy or other targeted therapies, emphasizing the necessity of developing of personalized therapy[14]. CD39, also known as ectonucleoside triphosphate diphosphohydrolase-1, is a transmembrane protein encoded by the ENTPD1 gene. It can transform ATP and ADP into immunosuppressive adenosine (ADO) through a cascade reaction under the synergistic action of a 5′-nucleotide enzyme (CD73)[15].

In the present study, via integrated analysis of TCGA BLCA bulk-seq, single-cell transcriptome and Imvigor210 RNA-seq datasets, we find that CD39 is significantly overexpressed in BC tissues and is associated with immunosuppression. Patients with higher CD39 levels show less potential to get benefit from aPD-L1 treatment. With a subcutaneous model, we demonstrate that inhibition of CD39 (CD39i) by sodium polyoxotungstate (POM-1, a novel ENTPDase inhibitor)[16] can inhibit BC growth and improve prognosis in vivo, which is associated with increased infiltration levels of CD8 + T cells and conventional type 1 dendritic cells (cDC1) in cancer tissues. In addition, we also find that CD39i has a synergistic effect with CIS, a chemotherapeutic drug that is widely used in the clinical treatment of BC, rather than in combination with immunotherapy agents (aPD1 or aPD-L1). Collectively, these results suggest the possibility of CD39i as a treatment target for BC, and the combination of CD39i and CIS may be the preferred treatment.

## Results

### CD39 is overexpressed in BC and mainly localized at the tumor stromal region

By analyzing our previously published data (8 tumor samples and 3 paratumor samples from BC patients)[14], we found that numerous immune checkpoints were prominently increased in tumor-derived lymphocytes (Supplementary Fig. 1A). We then analyzed TCGA-BLCA data using GEPIA[17], an online tool available at http://gepia.cancer-pku.cn/, and found that only CD39 was significantly associated with patient progression among all these immune checkpoints (Supplementary Fig. 1B). Therefore, CD39 was identified as important and the focus of this study.

After quality control and batch effect removal, all the single cells from 8 BC samples and 3 paratumor samples[14] were clustered into 10 major clusters based on known cellular markers (Fig. 1A, Supplementary Fig. 1C and Supplementary Fig. 2A), including epithelial cells, endothelial cells, smooth muscle cells and pericytes (SMC and peri), fibroblasts, myeloid cells, B cells, T and NK cells, mast cells, LYVE1 + cells, and plasma cells. To investigate the abundance of infiltrating immune cells in cancerous and paracancerous samples, we normalized the number of cells in each sample, taking 2000 single cells randomly from the data in each sample, then the single cells from 8 BC and 3 paracancer tissues were clustered into 10 major clusters (Supplementary Fig. 2B). The Supplementary Figure 2C showed the clustering diagram of 22,000 single cells collected randomly from the 11 samples. The results suggested that there were more immune cells in the paracancerous tissues. Using GEPIA2 (http://gepia2.cancer-pku.cn/#index)[18], the more immune cells in the paracancerous tissues were also present in the TCGA-BLCA data (Supplementary Fig. 2D, E), which is consistent with our results.

We visualized the expression distribution of CD39, the results suggested that CD39 was mainly expressed in endothelial cells, SMC and peri, myeloid cells, fibroblasts as well as lymphocytes (Fig. 1B, Supplementary Fig. 2F and Supplementary Fig. 3A). Although CD39 was expressed in paracancerous stromal and endothelial cells, the levels were lower than that in tumor-derived cells (Supplementary Fig. 2F). Immunohistochemical (IHC) staining of CD39 in BC and paracancer tissues was performed in a tissue array and revealed that CD39 expression in BC tissues was remarkably higher than that in normal bladder tissues ($P < 0.001$). In addition, BC patients with higher CD39 expression had a worse outcome (Fig. 1C–E and Supplementary Fig. 3B, C).

### Increased CD39 is associated with T-cell exhaustion and a limited ICB response rate

Previous studies have demonstrated that upregulation of CD39 on the surface of T cells indicates the terminal stage of T-cell dysfuncion[19,20]. With single-cell transcriptome, we found that tumor infiltrated T cells, whether CD4 + or CD8 + , both showed a significantly higher levels of CD39, along with increased LAG3, another marker of T-cell exhaustion (Fig. 1F, G and Supplementary Fig. 4A). Next, we visualized CD39, LAG3, CK5, and CK8 as individual samples with equal cell count. The results suggested that significant upregulation of T and NK cells' surface CD39 and LAG3 expression in the tumor, which did not occur in all tumor samples but was observed in most tumor samples (Supplementary Fig. 3A and Supplementary Fig. 4B). In the TCGA BLCA dataset, the CD39 expression level was positively correlated with the T-cell exhaustion signature ($P < 0.0001$) (Fig. 1H). Through reanalysis of the Imvigor210 RNA-seq dataset[21], we found that the expression level of PD-L1 differs between 7 molecular subtypes of BC. Although the MS2a.2 subgroup, which had the highest PD-L1 level, showed the highest CR or PR rate, the PD-L1 level could not explain the limited response rates of MS1a and MS2b1(Fig. 1I, J and Supplementary Fig. 5A). Interestingly, both MS1a and MS2b1 had noteworthily higher levels of CD39 than all other subgroups, indicating that the increase in CD39 may limit the effect of aPD-L1 agents.

Using a tissue array containing BC and paracancer tissues, we demonstrated that BC tumor tissues contain many more T cells than paratumoral mucosa at the protein level (Fig. 1K and Supplementary Fig. 5B). In addition, the expression level of CD39 was positively correlated with the proportion of exhausted CD8 + T cells, but not the abundance of CD8 + T cells (Fig. 1L, M), meaning that CD39 is only related to the degree of T-cell dysfunction and is not associated with the immune infiltration level of BC.

### CD39i suppresses BC progression in vivo

To verify the carcinogenic effect of CD39, we treated the MB49 cell-C57BL/6 J mouse subcutaneous tumor model and mouse bladder orthotopic tumor model with POM-1. The results showed that CD39i significantly inhibited the tumor growth and improved the survival rate of mice (Fig. 2A–C and Supplementary Fig. 5C). In a mouse orthotopic bladder cancer model, we found that treatment with CD39i could significantly reduce the maximum cross-sectional area of the tumor and the bladder weight of mice, again proving that treatment with CD39i could significantly inhibit tumor growth (Fig. 2D–G). The flow analysis results are shown in Fig. 2H–L and Supplementary Fig. 6A-C, CD39i treatment induced a significant increase in immune cell infiltration in the tumor tissues, including CD45 + cells, CD4 + and CD8 + T cells. To understand the panoramic change in tumor infiltrated immune cells after CD39i, we sorted the CD45 + cells from the POM-1- and PBS-treated groups and then performed single-cell RNA sequencing (10X Genomics). After quality control, using cell-type-specific gene markers (Supplementary Fig. 7), 18547 high quality cells were clustered into 9 major cell types, including T cells, myeloid cells, neutrophils, dendritic cells (DCs), NK cells, B cells, mast cells, Ctsk+ cells, and malignant cells (Fig. 2M). However, we failed to find any drastic change in the major components of CD45 + cells between the CD39i and control groups. Only a slight increase in the proportion of T cells was observed (Fig. 2N). Combined with the results of flow cytometry, all these results indicate that CD39i may be able to increase

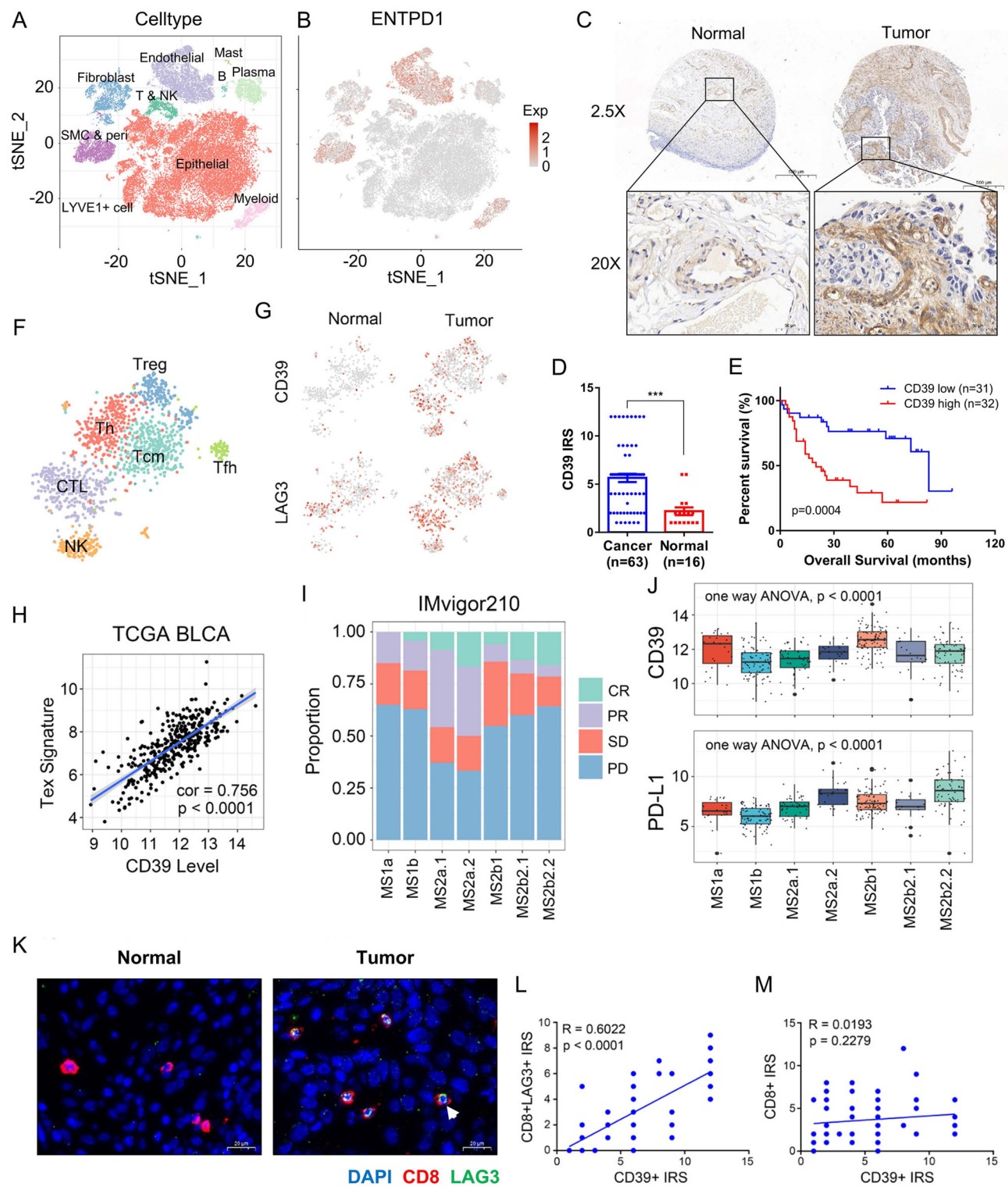

the abundance of all immune cell types, converting "cold" tumor into "hot" tumor to a certain extent.

## CD39i activated expansion of cDC1

According to classical markers[22], reclustering of DCs identified 6 subgroups, including migratory DC2 (mDC2), migratory DC1−CD80 + (mDC1−*CD80* + ), cDC1, cDC1-proliferating (cDC1-p), conventional type 2 dendritic cells (cDC2), and plasmacytoid DC (pDC) (Fig. 3A−C). cDC1-p refers to the cDC1 in G2M or S phase (*Mki67* + /*Mcm2* + ), accounting for a considerable portion of cDC1. Compared with the control group,

DC subpopulations from the CD39i treated tumors showed dramatic changes, embodying in decreased mDC1−CD80 + and pDC, along with significant increases in the cDC1 and cDC1-p proportions (Fig. 3D). However, although the proportion of DC subgroups has been relatively remodeled, we did not observe any remarkable changes in functional genes in each cell subgroup after CD39i treatment (Fig. 3E−G). These results indicated that CD39i only increases the proportion of cDC1 and promotes cDC1 expansion, but does not directly influence the function of DCs. For the monocyte-macrophage lineage, although much higher heterogeneity was observed, these cell subgroups were not influenced

**Fig. 1 | Expression distribution and function of CD39. A** The single cells from 8 BC and 3 paracancer tissues were clustered into 10 major clusters. **B** The CD39 was mainly expressed in endothelial cells, smooth muscle cells, pericytes, myeloid cells, fibroblasts, and lymphocytes. **C** Immunohistochemical staining of CD39 in BC ($n = 63$) and paracancer ($n = 16$) tissues in a tissue array. Scale bars, 500 μm (upper) and 50 μm (lower). **D** The expression levels of CD39 staining in BC (Mean ± SEM: 5.651 ± 0.4354, $n = 63$) and paracancer (Mean ± SEM: 2.188 ± 0.4105, $n = 16$) tissues were assessed using Remmele and Stegner's semiquantitative immunoreactive score (IRS) scale. The two-side unpaired Student's *t*-test was used for two-group comparisons of values. **E** A high level of CD39 predicted poor prognosis in 63 BC patients. The log-rank (Mantel–Cox) test was used to compare the survival curves. **F, G** The differences in CD39 expression patterns between lymphocytes of BC ($n = 8$) and paracancer ($n = 3$) tissue origin, and the correlation between CD39 and LAG3. **H** Correlation between the CD39 expression level and T-cell exhaustion in the TCGA-BLCA cohort. A linear regression model (Pearson's correlation) was used to determine the individual correlations between different variables. **I, J** Correlation between the clinical outcome of aPD-L1 agent (atezolizumab) and the expression level of PD-L1 or CD39 in 7 molecular types of BC patients reported by Mariathasan et al.[21], MS1a: $n = 23$, MS1b: $n = 79$, MS2a1: $n = 45$, MS2a2: $n = 25$, MS2b1: $n = 92$, MS2b2.1: $n = 18$, MS2b2.2: $n = 66$. Bounds of the box spans from 25% percentile to 75% percentile, a dashed line shows median, and whiskers indicate minima and maxima. **K** Immunofluorescence staining (DAPI: blue, CD8: red, LAG3: green) of exhausted T cells in BC ($n = 63$) and paracancer ($n = 16$) tissues in a tissue array, Mean ± SEM of CD8 + LAG3 + IRS: 2.778 ± 0.2971, $n = 63$, Mean ± SEM of CD8 + IRS: 3.698 ± 0.3108, $n = 63$. Scale bars = 20 μm. **L, M** Correlation between the expression level of CD39 and the proportion of T-cell exhaustion (Mean ± SEM: 2.778 ± 0.2971, $n = 63$) or CD8 + T-cell infiltration (Mean ± SEM: 3.698 ± 0.3108, $n = 63$). A linear regression model (Pearson's correlation) was used to determine the individual correlations between different variables. Source data are provided as a Source Data file (**D, E, L, M**). *P*-values <0.05 were considered significant: \*$P < 0.05$; \*\*$P < 0.01$; \*\*\*$P < 0.001$; \*\*\*\*$P < 0.0001$.

after CD39i treatment, nor were the proportions of cell subgroups or functional genes (Fig. 3H–J).

## CD39i treatment activated CD8 + T cells and promoted T-cell expansion

To further investigate CD39i-mediated changes in T cells, the lymphocytes were clustered into 10 clusters, including CD8-G2M, CD8-S, CD8-Slfn5, CD8-TRM, CD8-Ifitm1, NK, Th, CD4-Pdcd1, Th17, and Treg (Fig. 4A–C). As shown in Fig. 4D, we found that an increase in cycling CD8 + T cells (CD8-G2M and CD8-S) was the most significant change caused by CD39i. We also observed a slight decrease in Treg abundance and a slight increase in the proportion of NK cells, which was confirmed by flow cytometry (Fig. 4E, F and Supplementary Fig. 8A). Considering that CD39i-mediated increase in the absolute count of immune cells (Fig. 2H, I), and CD39i still fetches considerable enrichment in NK cell abundance, which has been confirmed to have a role in the treatment of BC[23,24]. Subsequently, we performed a basic analysis of the NK cell phenotype through flow cytometry and found that CD39i treatment significantly increased the proportion of CD27 + CD11b + and CD27-CD11b + NK cells, but decreased the proportion of CD27-CD11b- and CD27 + CD11b- NK cells, indicating an increase in mature NK cells[25] (Supplementary Fig. 8B, C). However, we found no difference in the proportion of Granzyme B + NK cells between control and CD39i treated group (Supplementary Fig. 8D, E), suggesting the differences in the percentage of Granzyme B + NK cells appear to be driven entirely by increased NK cell numbers and not increased Granzyme B expression/activation on a per cell basis. The same applies to the interpretation of XCL1 secretion. By analyzing the data of scRNA-seq data, we found that CD39i treatment did not change the expression level of XCL1 on a per cell basis (Supplementary Fig. 8F).

In addition, we examined functional changes in different CD8 + T-cell subpopulations. The results demonstrated that after treatment with CD39i, the common tissue resident markers (*Gzmk* and *Cd69*) of cycling CD8 + T cells were downregulated, while *Eomes* and *Prf1*, along with a series of granzymes (*Gzmb, Gzmc, Gzme, Gzmf*), were significantly upregulated (Fig. 4G), indicating enhanced cytotoxicity. Moreover, we found that CD8-TRM cells, not expanding T cells, showed the highest expression levels of *Gzmb* and *Gzma*. In contrast, the interferon induced subgroups, CD8-Slfn5 and CD8-Ifitm1 expressed the highest levels of *Ifng* and *Tnf*. All of these genes have been commonly used to assess CD8 cytotoxicity (Fig. 4G). Through flow cytometry analysis, we found that CD39i treatment promoted T-cell proliferation (Supplementary Fig. 9A, B) and enhanced T-cell tumor killing function by secreting more granzyme B and perforin, but not IFN-γ (Supplementary Fig. 9C-H). Interestingly, we also observed that co-stimulatory and co-inhibitory checkpoints were almost absent in CD8 + T cells except for the cycling cells (Fig. 4G), indicating that CD39i treatment did not completely reverse the tumor

immunosuppressive microenvironment, that is, while CD39i enhances the cytotoxicity of cycling CD8 + cells, the anticancer function of these cells might also inhibited by other negative regulatory mechanisms, such as PD1/PD-L1 pathway. Taken together, these results demonstrated that CD39i activated the cytotoxic T cells (CTLs) and promoted the expansion of these cells.

Previous studies have confirmed that cDCs, especially cCD1, have an essential regulatory effect on CTLs[26–28]. To investigate whether CD39i influences the communication network between NK cells, cDC1 and T cells, we generated a cell–cell communication network in the subcutaneous tumor with CellPhoneDB 2[29,30], a Python-based database of cell receptors, ligands, and their interactions that can be used to study cell-to-cell interactions at the molecular level. We found that cDC1 and mDC1 expressed Il15, Il12b, Cd80, and Cd86, which might enhance the antitumor activity of cycling CD8 + T cells by interacting with Il15ra, Il12rb1/2, and Cd28 (Fig. 4H). Previous studies have shown that Il12b-secreting mDC are mainly derived from cDC1[22,31]. NK cells communicate with cDC1 by expressing Xcl1, which leads to the chemotaxis of cDC1 into the tumor, and CD39i can enhance this effect by increasing NK cells in the tumor. Additionally, we also observed that cDC1 and mDC1 could potentially negatively regulate the function of cycling CD8 + T cells by expressing CD274 (Pd-l1), Lgals9, and Tnfsf9 (Fig. 4H). Collectively, our results suggested that CD39i treatment increased NK cell infiltration, and then released more Xcl1 (in total rather than on a per cell basis) to recruit cDC1, which activated cycling CD8 + T cells by expressing Il15, Il12b, Cd80 and Cd86 when they matured, thus achieving an antitumor effect (Fig. 4I). Moreover, this process is negatively regulated by immune checkpoints.

## Depletion of NK cells reverses the antitumor effects of CD39i in vivo

To verify the potential mechanism by which CD39i inhibits the progression of BC, we constructed a mouse subcutaneous tumor model again and randomly divided the mice into four groups. The mice were treated with isotype control antibodies, CD39i, aNK1.1, and CD39i + aNK1.1, respectively. As shown in Fig. 5A–C and Supplementary Fig. 10A, the results again showed that CD39i significantly inhibited the tumor growth and improved the overall survival rate of mice compared with the control group ($P < 0.0001$), but the effect of CD39i was significantly reversed in the NK cells-depleted mice ($P < 0.0001$). Supplementary Fig. 10B, C shows that the aNK1.1 antibody significantly reduced NK cells in tumors. Additionally, we found that CD39i treatment increased the proportion of tumor infiltrated cDC1 (Fig. 5D and Supplementary Fig. 10D, E), XCR1 + cDC1 (Supplementary Fig. 10F, G) as well as the tumor Xcl1 protein levels (Supplementary Fig. 10H) in control mice, but not in NK cell absent mice. The above results indicated that NK cells were the key for CD39i to exert the antitumor effect.

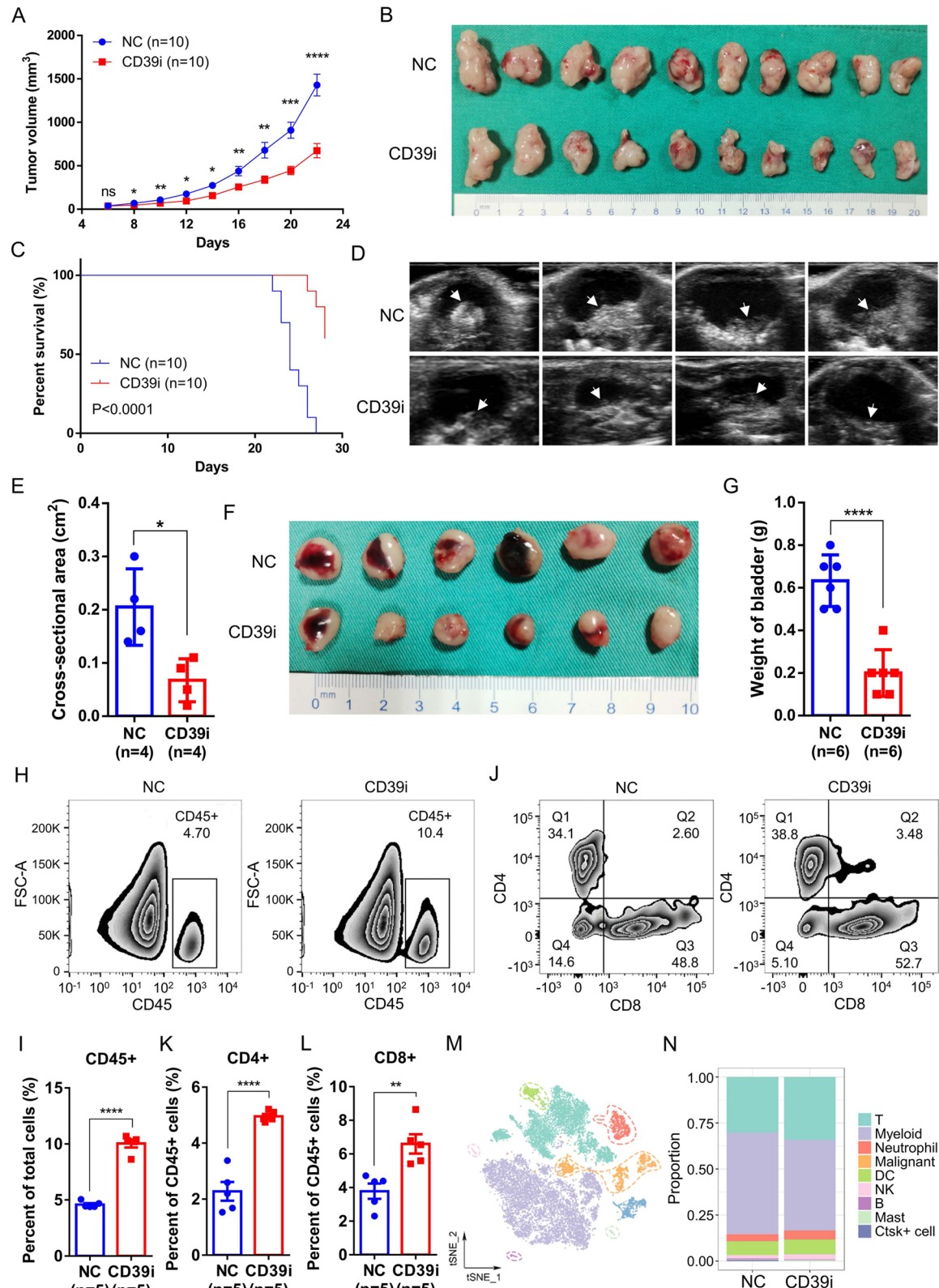

## cDC1-deficient leads to therapeutic failure of CD39i with a higher tumor burden

To elucidate whether cDC1 has a key role in CD39i-mediated antitumor effects in BC, we established a *Batf3*[−/−] cDC1-deficient model[22]. As described before, tumor models were constructed by subcutaneous injection of MB49 cells into *Batf3*[−/−] or *Batf3*[+/+] (WT) C57BL/6 J mice, and then treated with POM-1 or PBS, respectively. The results again

demonstrated that CD39i could effectively inhibit tumor growth and improve the prognosis of tumor-bearing *Batf3*[+/+] mice. In contrast, the *Batf3*[−/−] mice did not benefit from CD39i, as CD39i failed to inhibit the growth of subcutaneous tumors or improve the prognosis of *Batf3*[−/−] mice (Fig. 5E–G and Supplementary Fig. 11A). The difference in BATF3 expression between *Batf3*[+/+] and *Batf3*[−/−] cDC1-deficient mice tumors was confirmed by IHC (Supplementary Fig. 11B).

**Fig. 2 | CD39i suppressed BC progression by increasing the CD45 + immune cell populations in tumor tissues.** CD39i noteworthily inhibited the subcutaneous tumor growth (**A**, **B**) and improved the survival rate of mice (**C**), $n = 10$ for each group. The two-side unpaired Student's $t$-test was used for the tumor size comparison of different groups at the same time point. The log-rank (Mantel–Cox) test was used to compare the survival curves. CD39i noteworthily reduced the maximum cross-sectional area (Mean ± SEM: 0.2050 ± 0.0359 vs. 0.0675 ± 0.0202) of the orthotopic mouse bladder tumor, $n = 4$ for each group (**D**, **E**) and the bladder weight (Mean ± SEM: 0.6333 ± 0.0494 vs. 0.2000 ± 0.0447) of mice, $n = 6$ for each group (**F**, **G**). The two-side unpaired Student's $t$-test was used for two-group comparisons of values. Flow cytometry analysis showed that CD39i treatment induced a significant increase in immune cell infiltration within the tumor tissues, including CD45 + cells (**H**, **I**), CD4 + and CD8 + T cells (**J**–**L**). Mean ± SEM: I 4.598 ± 0.1262 vs. 10.05 ± 0.3638, K 2.277 ± 0.3334 vs. 4.958 ± 0.0859, L 3.782 ± 0.4524 vs. 6.598 ± 0.5756. All the flow cytometry analyses were repeated 3 times with 5 samples in each group. **M** The single cells from subcutaneous tumors (3 subcutaneous tumors from 3 mice mixed 1:1:1) of the control and CD39i groups were clustered into 9 major cell types. **N** There was no difference in the proportion of the cell clusters between the control and CD39i groups. Source data are provided as a Source Data file (**A**, **C**, **E**, **G**, **I**, **K**, **L**). $P$-values <0.05 were considered significant: *$P < 0.05$; **$P < 0.01$; ***$P < 0.001$; ****$P < 0.0001$.

To parse the effect of *Batf3*[−/−] cDC1-deficient on the tumor microenvironment and CD39i efficacy, we performed single-cell sequencing on WT and *Batf3*[−/−] mice treated with POM-1 or PBS. The single cells were clustered into 10 major clusters, including myeloid cells, T cells, B cells, neutrophils, dendritic cells (DCs), NK cells, mast cells, Ctsk+ cells, stromal cells, and epithelial cells, all of which were captured in four different samples (Fig. 6A, B). Similar to the first batch, CD39i caused significant enrichment and expansion of cDC1, while this effect was completely blocked in the *Batf3*[−/−] model (Fig. 6C, D). Notably, *Batf3* deficiency did not influence the proportion of cDC2. Instead, the proportion of mDC was significantly increased (Fig. 6D). Compared with WT mice, mDC from *Batf3*[−/−] mice expressed more Ccl17 and Ccl22 and less Il12b (Fig. 6E), indicating that these cells were mainly derived from cDC2[22,31]. These results also imply that *Batf3*[−/−] may promote the maturation of cDC2, as the mDC/cDC2 ratio increased in *Batf3*[−/−] mice.

With these gene markers (Supplementary Fig. 11C), the lymphocytes were clustered into 15 clusters, including CD4-C01-Tcm, CD4-C02-Th, CD4-C03-Th-PD1, CD4-C04-Th17, CD4-C05-Treg, CD8-C01-TRM, CD8-C02-Ifit, CD8-C03-Tex, CD8-C04-Tex, CD8-C05-Cycling, NK-C01, NK-C02, NK-C03-Gzmf, NK-C04, and NK-C05-Cycling (Fig. 6F). We observed that CD39i treatment significantly increased the proportions of proliferating CD8 + T cells and 5 subtypes of NK cells, and slightly decreased the Treg abundance in WT mice, as observed in the first batch (Fig. 6G). CD39i treatment increased NK cell numbers in *Batf3*[−/−] cDC1-deficient mice, but the effects on CD8 + T cells were ablated (Fig. 6G), indicating that CD39i indirectly activates T cells through the NK/cDC1 axis. Additionally, we also found that *Batf3* deficiency caused an obvious increase in Treg abundance, consistent with previous studies[32,33]. For the monocyte-macrophage lineage, both CD39i and *Batf3* deficiency failed to influence these cells, which indicated that this lineage may not be involved in the CD39i-mediated antitumor effect (Supplementary Fig. 11D and Fig. 6H, I).

**Investigation of CD39i-based combination therapy**

As shown in Fig. 4G, the expression levels of CD39 and several co-inhibitory receptors (PD1, LAG3, and TIM3) were upregulated following the activation of CD8 + T cells, which may be negative regulate the cytotoxicity of CTL. Since then, combination treatment with aPD1 or aPD-L1 might be a better choice. However, subsequent in vivo experiments demonstrated that although monotherapy with CD39i or aPD-L1 antibody showed similar efficacy in inhibiting the tumor growth and improving the survival rate of the MB49 cell-subcutaneous tumor model, the combination therapy of CD39i and aPD-L1 did not show any synergistic effect (Supplementary Fig. 12A and Fig. 7A–C). Through flow cytometry analysis, we found that CD39i treatment significantly reduced the number of precursor exhausted T cells (Supplementary Fig. 12B, C), while aPD-1 is thought to enhance the activation/expansion of precursor exhausted T cells[34], which may undermine their potential to act synergistically.

Next, we tried different combination strategies of CD39i, aPD1 and CIS. As shown in the Supplementary Fig. 13A and Fig. 7D–F, mono-therapy with CD39i, aPD-1, or CIS remarkably inhibited tumor growth and improved the prognosis. For combination strategies, CD39i + CIS and aPD1 + CIS, rather than CD39i + aPD1, had significant synergistic effects on inhibiting tumor growth and improving prognosis. In addition, we found that the combination of CD39i, CIS, and aPD1 had the strongest antitumor efficacy in vivo. To explore the synergistic mechanism of the combination of CD39i and CIS, flow cytometry analysis was performed. We found that cisplatin at doses ≥ 3 mg/kg had a synergistic effect with CD39i on increasing the proportion of cDC1 in tumors, but there was no difference in the synergistic effect of cisplatin at 3 mg/kg and 4 mg/kg (Supplementary Fig. 13B, C). Further, we found that CD39i (but not 3 mg/kg cisplatin alone) treatment significantly increased the proportion of tumor infiltrated cDC1 cells. And cisplatin at 3 mg/kg had a synergistic effect with CD39i on increasing the proportion of cDC1 in tumors (Fig. 7G and Supplementary Fig. 13D). Considering that the cytotoxicity of cisplatin can cause tumor cell death and release large amounts of ATP[35], and that the use of CD39i can inhibit the degradation of extracellular ATP[16,36], we speculate that the combined regimen of the two may further increase the accumulation of ATP in the tumor microenvironment and then mediate a stronger antitumor effect by continuously activating and enhancing the function of immune cells. Since CIS is the most commonly used chemotherapy drug at present, CD39i or a combination of CD39i and aPD-1 could be the follow-up treatment post chemotherapy.

## Discussion

In recent decades, local tumor resection combined with intravesical BCG immunotherapy has been the first-line treatment for NMIBC, but the overall therapeutic effect has been unsatisfactory, as a substantial proportion of patients still experience disease recurrence and progression[9,10]. Similarly, CIS-based chemotherapy combined with radical cystectomy has not been satisfactory as a recommended treatment for MIBC and advanced BC[37,38], and ~50% of patients with MIBC are ineligible for CIS-based neoadjuvant chemotherapy because of age-related and/or disease-related risk comorbidities[11]. The use of immune checkpoint inhibitors is a newly emerging therapeutic strategy to address these deficiencies. Several PD-1 inhibitors and PD-L1 inhibitors are already approved to treat MIBC and platinum-resistant metastatic urothelial cancer[37,39,40], but the reported objective response rates only range from only 13–25%[37]. Therefore, it is urgent to develop new therapeutic targets or combined therapy strategies for BC.

It is well known that ATP release from the intracellular to the extracellular space during necrosis, apoptosis and mechanical injury acts as a "danger" signal and activates the immune system by binding to P2 receptors[41–44]. In contrast, CD39 plays the role of a rate-limiting enzyme in hydrolyzing ATP and ADP into AMP, and AMP is then converted by CD73 to immunosuppressive adenosine, playing a series of biological roles by binding to P1 receptors[15]. A recent study[45] confirmed that the high expression of CD39 is involved in bladder tumorigenesis and is correlated with the early stage of bladder cancer by immunohistochemistry. Another study[46] revealed that the high expression of CD39 tended to result in a worse survival rate, and the elevated expression of CD39 in CD4 +/CD8 + T cells was significantly associated with the pathological T stage. Currently, the ATP-adenosine

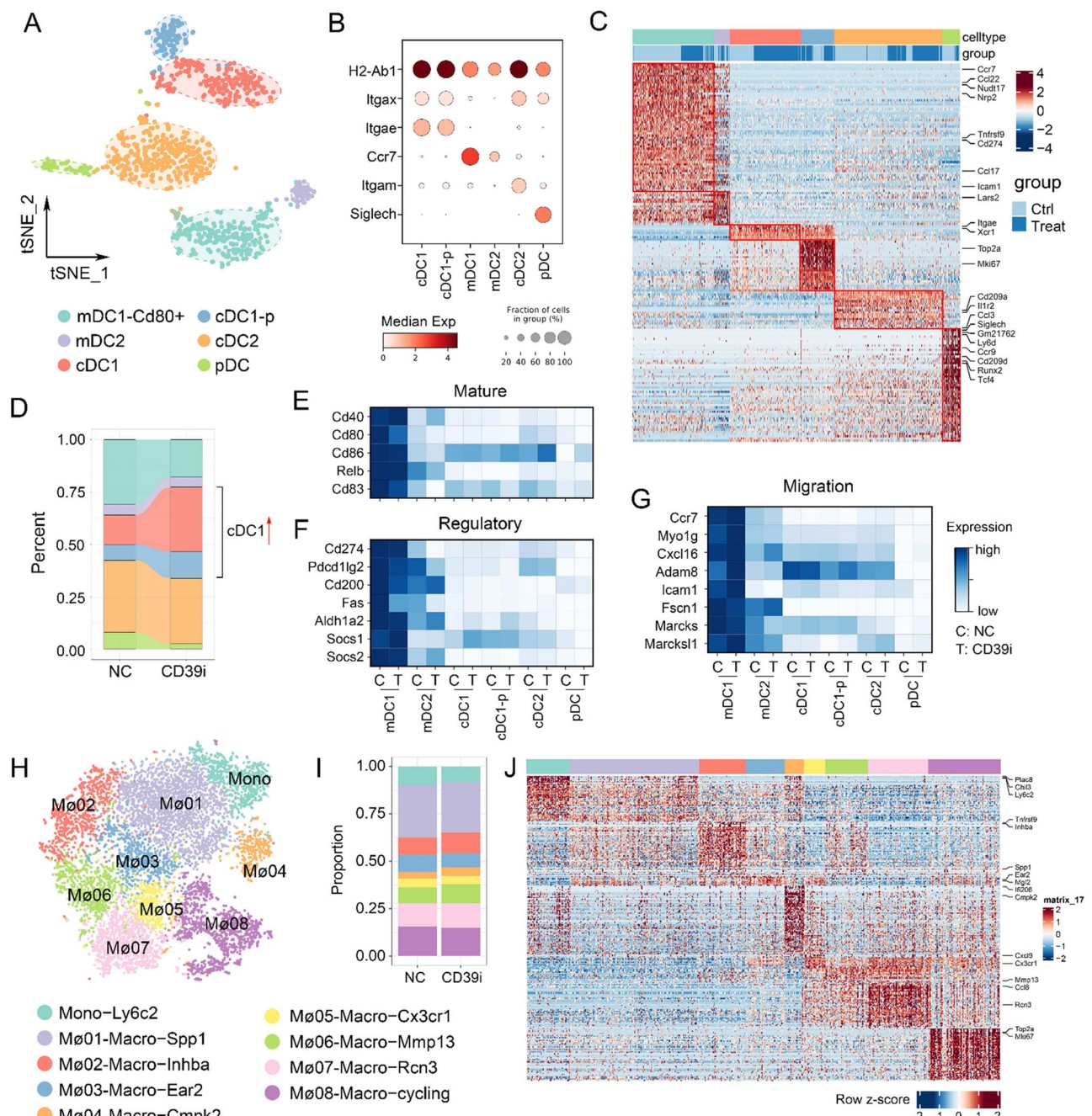

**Fig. 3 | The proportion of DC subpopulations and mononuclear macrophage subpopulations changed after CD39i treatment. A**, **B** The DCs were clustered into 6 clusters according to classical markers. **C** The expression of function-related genes in different DC subpopulations from the control group and the CD39i treatment group. **D** The proportion of DC subpopulations derived from the control group and CD39i treatment group (3 subcutaneous tumors from 3 mice mixed 1:1:1). **E**–**G** There were no differences in mature, regulatory, and migration related functional gene expression after CD39i treatment. **H**–**J**. The mononuclear macrophages showed greater heterogeneity and a large number of different subpopulations, but the proportion of each subpopulation and the functional gene expression levels were not noteworthily changed after CD39i treatment.

immune checkpoint pathway is viewed as a valuable target for cancer therapy, and many inhibitors or monoclonal antibodies have entered the clinical trial stage[15,41].

In this study, through single-cell RNA-sequencing analysis of bladder cancer tissue samples, we found that tumor infiltrated CD8 + T cells and NK cells showed a significantly higher levels of CD39, along with increased T-cell exhaustion markers. In the TCGA-BLCA dataset, the CD39 expression level was positively correlated with the T-cell exhaustion signature. The high expression of CD39 could partly explain the low response rate of ICB that could not be explained by PD-

L1 expression. Furthermore, by using a tissue array containing of BC and paracancer tissues, we found that the expression level of CD39 was positively correlated with the proportion of exhausted CD8 + T cells, but was not related to the abundance of CD8 + T cells, indicating that CD39 might play an important role in the regulation of the tumor microenvironment, and that targeting CD39 may be an effective strategy for the treatment of BC.

Previous studies have found that CD39i facilitated the significant expansion of intratumor effector T cells and rescued aPD-1 resistance by triggering an eATP-P2X7-inflammasome-IL18 axis[44]. Our work

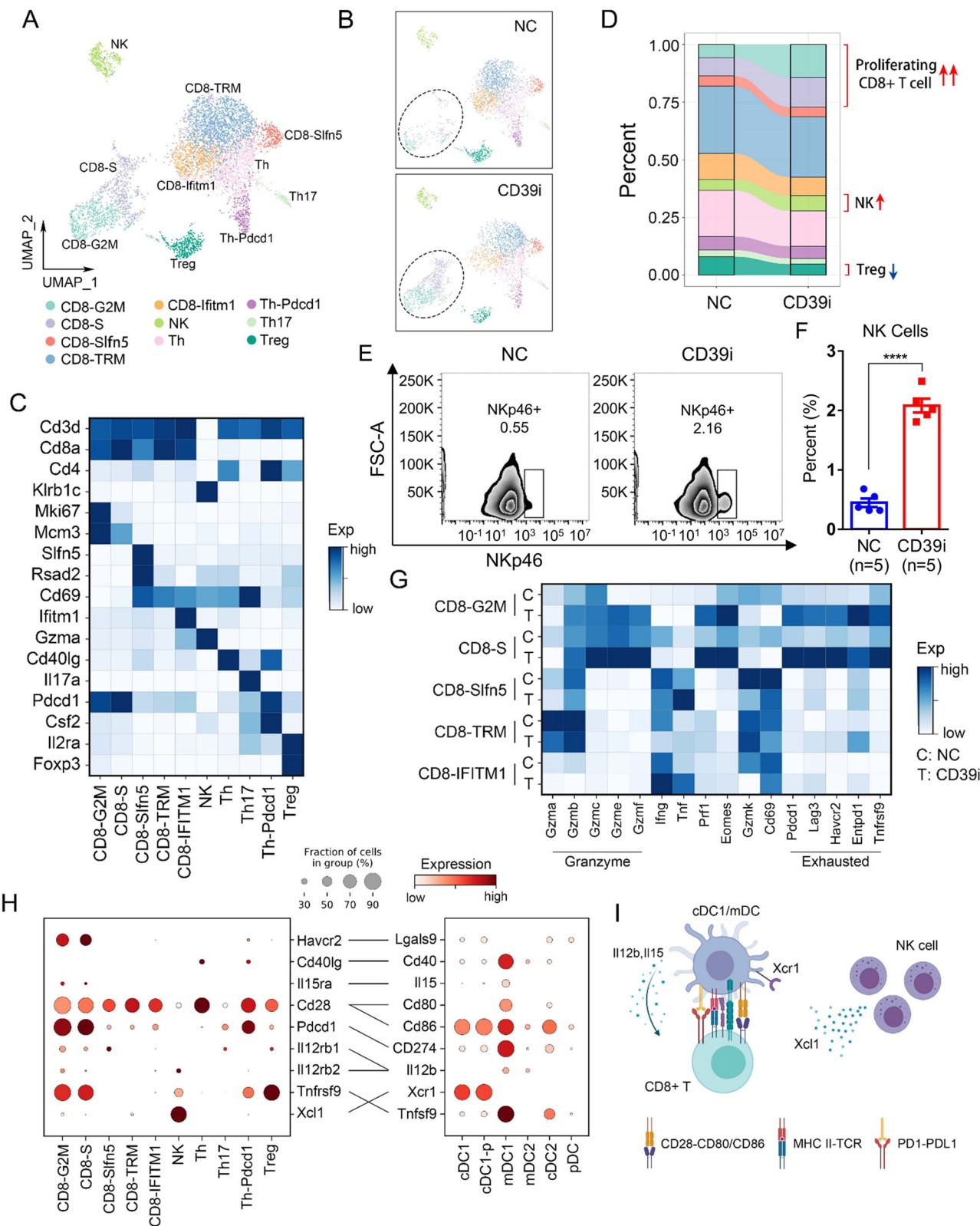

provided a panoramic view of CD39i-mediated changes in the tumor microenvironment at the single-cell level, that is, CD39i noteworthily inhibited the tumor growth and improved the survival rate of mice by increasing the intratumor NK cells, cDC1 subpopulations, and CD8 + T cells, but decreasing Tregs. According to a recent study[47], NK cells stimulate the recruitment of cDC1 into the tumor microenvironment, promoting cancer immune control. We propose that CD39i inhibits BC by increasing the number of intratumor NK cells and then

recruiting more cDC1 to activate CD8 + T cells through the cell–cell communication network. The NK cells, cDC1, and CD8 + T cells together form a regulatory loop, promoting each other. However, this loop was blocked in both the NK cell absent mice and the cDC1-deficient *Batf3*[−/−] mice, suggesting that NK cells and cDC1 play critical roles in this process, and CD39i indirectly activates T cells through the NK/ cDC1 axis. These results contribute to enhancing our understanding of the role of CD39i and point to new directions for future research.

**Fig. 4 | The proportion of lymphocyte subpopulations changed after CD39i treatment. A, B** The lymphocytes derived from the control group and the CD39i treatment group were clustered into 10 clusters according to classical markers. **C** The expression of classical markers in different lymphocyte subpopulations. **D** The proportions of different lymphocyte subpopulations in the control group and the CD39i treatment group (3 subcutaneous tumors from 3 mice mixed 1:1:1). **E, F** Flow cytometry showed that CD39i treatment resulted in the upregulation of NK cells (CD45 + CD3-NKP46 + ), Mean ± SEM: 0.4480 ± 0.0718 vs. 2.082 ± 0.1169. The two-side unpaired Student's t-test was used for two-group comparisons of

values. The flow cytometry analyses were repeated 3 times with 5 samples in each group. Source data are provided as a Source Data file. **G** CD39i treatment resulted in enhanced function (cytotoxicity) of CD8 + T cells and upregulated expression of several inhibitory receptors (Pd1, Lag3, and Tim3). **H** The molecular interactions between CD45 + immune cell populations via specific protein complexes, predicted by CellPhoneDB 2[29, 30]. **I** A conceivable cell–cell communication network (Created with BioRender.com.) of immune cells in the subcutaneous tumor. P-values <0.05 were considered significant: *$P < 0.05$; **$P < 0.01$; ***$P < 0.001$; ****$P < 0.0001$.

CD8 + T cells are important for the protective immunity against intracellular pathogens and tumor[48]. However, the persistent encounters with antigens lead to exhaustion of CD8 + T cells[49]. Eomes-expressing CD8 + T cells had higher expression of several inhibitory receptors[50], consistent with more severe exhaustion[51]. In this study, we found that although CD39i greatly promoted CTL amplification and cytotoxicity, it was also accompanied by a remarkable increase in co-inhibitory receptors, indicating that the anticancer function of CD39i may be limited by other negative regulatory mechanisms. To seek for a better therapeutic effect, we tried to combine CD39i with the current first-line medication for BC. Unexpectedly, although CD39i, aPD1, and aPD-L1 alone significantly slowed the progression of BC, neither CD39i + aPD1 nor CD39i + aPD-L1 had synergistic effects. This was quite different from what has been observed in other tumors, where multiple studies have demonstrated the synergistic effects of CD39i and aPD1 antibodies in the melanoma, renal carcinoma, mammary carcinoma, and colon adenocarcinoma tumor models[44,52]. After flow cytometry analysis, we found that CD39i treatment conspicuously reduced the number of precursor exhausted T cells, a population that was thought to be susceptible to PD-1 blockade-mediated activation/expansion[34]. Therefore, we speculate that both CD39i and aPD-L1 treatments may exert antitumor effects by affecting the same cell population, which may undermine their potential to act synergistically. Delightfully, a significant synergistic effect was observed between CD39i and CIS or the combination of aPD1 and CIS, and the efficacy of the three-drug combination strategy in inhibiting the progression of BC was prominently superior to that of the combination of the two agents. Through flow cytometry analysis, we found that cisplatin had a synergistic effect with CD39i on increasing the proportion of cDC1 in tumors. Based on the available evidence[42,44], we deduced that the cytotoxicity of CIS resulted in massive amounts of tumor cell death, which released large amounts of eATP and enhanced T-cell activation. Combined treatment with CD39i significantly slowed the degradation rate of eATP and maintained the immunologic activation state, thereby exerting a synergistic role in inhibiting tumor progression.

The effects of CD39i, as well as other means of immunotherapy, largely depend on the degree of immune cell infiltration, which is in line with current achievements in the field of bladder cancer immunotherapy. For immune desert type tumors, promoting immune infiltration by small molecule drugs can significantly reverse the problem of immunotherapy tolerance. For tumor tissues, the proportion of immune cells in bladder cancer depends on its molecular subtype, which was discussed in our previous paper[14]. Tumors of different molecular subtypes could vary in their degree of T-cell infiltration several- or even dozens of folds. Therefore, the large difference in the proportion of epithelial-immune cells among the 8 tumor samples might be a manifestation of the differences in tumor molecular typing. Generally, immune exclusion occurs in luminal-papillary type of bladder cancer with few immune cells and stromal cells (T4, T8), in this part of tumor tissue, which belongs to immune noninvasive type tumor, not immune depletion tumor. Meanwhile, current clinical studies have found that anti-PD-L1 therapy is only suitable for PD-L1 + patients, and for patients with immune exclusive, how to promote the entry of immune cells into the tumor remains a great challenge. Our study found that high CD39 expression was usually accompanied by

enrichment of CD8 + LAG3 + exhausted T cells, indicating the amount of CD39 expression was positively correlated with T-cell exhaustion. Meanwhile, through reanalysis of Imvigor210 RNA-seq dataset[21] we found that patients with moderate expression of PD-L1 and high expression of CD39 responded poorly to treatment with anti-PD-L1, which might be the preferred population for CD39i treatment strategies. Therefore, taken together with the results of in vivo experiments, we propose that CD39i treatment might be a better choice for patients with higher CD39 expression, especially in patients who fail to respond to anti-PD-1 or anti-PD-L1 therapy.

POM-1 is an inhibitor of ENTPDase and showed the most significant inhibition of CD39/ENTPDase1[36]. In this study, POM-1 was used to inhibit CD39, but previous studies showed that POM-1 also inhibited ENTPDase2 and ENTPDase3[16,36], which is a shortcoming of this study.

In conclusion, using single-cell sequencing, we provided a panoramic view of CD39i-mediated changes in the tumor microenvironment and enhanced our understanding of the role of CD39i. CD39i exerts an antitumor effect by increasing the proportion of tumor infiltrated NK cells, cDC1 and cycling CD8 + T cells. At the same time, our work has provided a combined regimen based on CD39i, laying a solid foundation for clinical transformation.

## Methods
### Cell culture
The mouse-derived BC cell line MB49 obtained from Otwo Biotech Inc. (cat #: HTX2716, Guangzhou, China) was cultured in RPMI 1640 medium (Servicebio Co., Ltd., China) supplemented with 10% fetal bovine serum (FBS, Gibco, USA) and 1% penicillin-streptomycin (Beijing Solarbio Science & Technology Co., Ltd., China) in a cell incubator (Thermo Fisher Scientific, Inc., USA) with a controlled atmosphere at 37 °C that was humidified and contained 5% $CO_2$. The cell line was tested negative for mycoplasma. The MB49 cell line was authenticated by STR profiling.

### Animals and drug administration
All animal procedures were approved by the Animal Care and Use Committee of Tongji Medical College of Huazhong University of Science and Technology (IACUC Number: 2580). Wild-type (WT) C57BL/6 J male mice (6–8 week old) purchased from GemPharmatech Co., Ltd., Nanjing, China and Batf3$^{-/-}$ male mice (6–8 week old) on a C57BL/6 mouse background purchased from the Shanghai Model Organisms Center Inc. were housed in a pathogen-free facility at 22 °C with 50% humidity and 12 h light/12 h dark cycles. For syngeneic tumor experiments, $5 \times 10^5$ syngeneic MB49 cells in 100 μl PBS were injected subcutaneously into the right armpits of age-matched mice (Day 0). The mouse orthotopic bladder cancer model was constructed based on a previous study[53]. In short, the mouse bladder mucosa was cauterized by a platinum wire electrode using a monopolar coagulation at 2 W for 1 s, and then $5 \times 10^5$ syngeneic MB49 cells in 100 μl PBS were instilled into the bladder for 2 h. Six days after tumor implantation, all the mice were randomized into groups of 10–14 mice. Each group was treated intraperitoneally with the following regimens. (1) The NTPDase inhibitor sodium polyoxotungstate (POM-1/CD39i, 5 mg/kg, purchased from Santa Cruz Biotechnology, Inc., Dallas, Texas, USA) was administered once per day from Day 6 to Day 22. (2). A 10 mg/kg

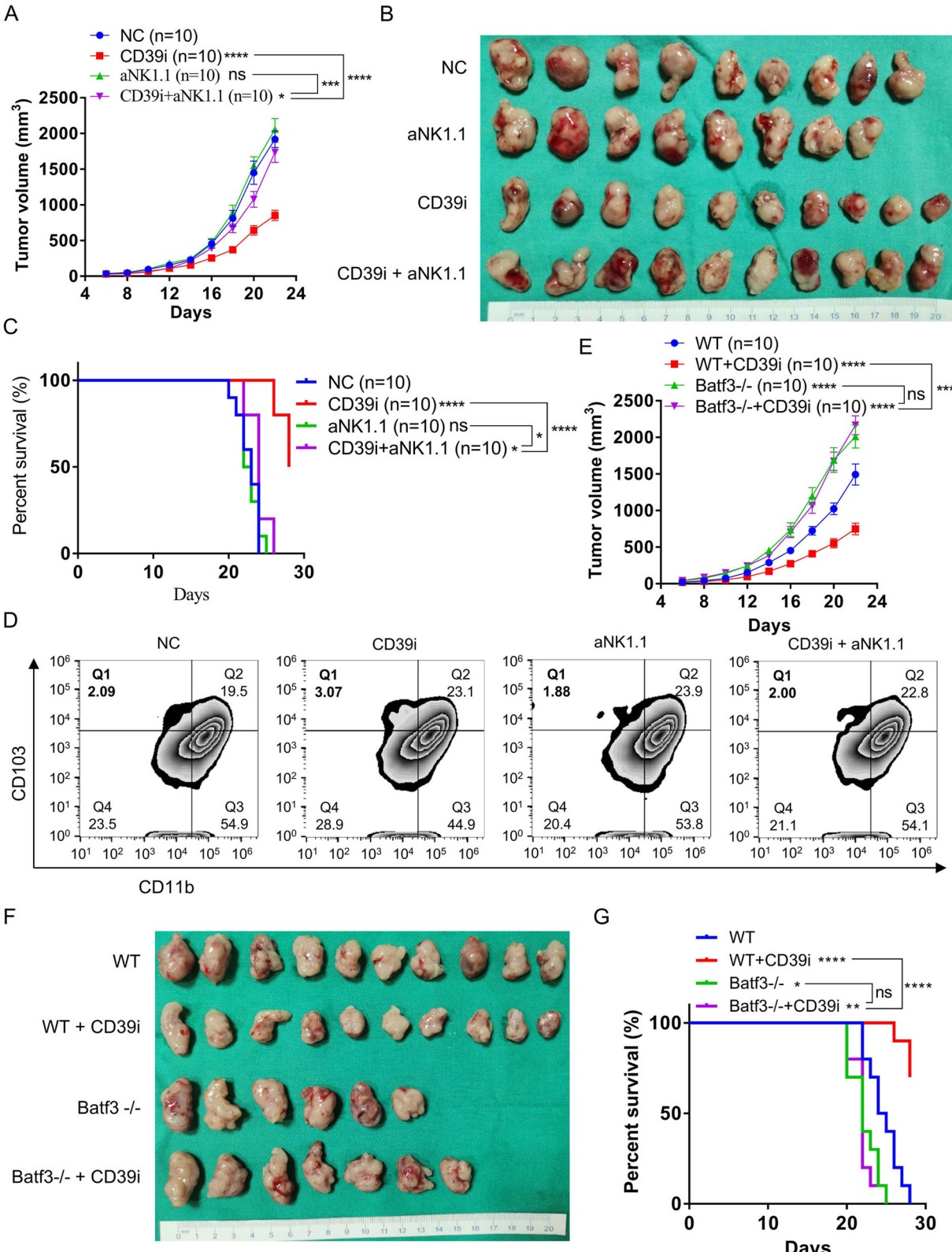

monoclonal anti-mouse NK1.1 antibody (aNK1.1, cat #: BE0036, Bio X cell, NH, USA) was administered on Days 6, 13, and 20 (once a week) to deplete NK cells in mice[54]. (3). Monoclonal anti-mouse PD-1 (CD279) antibody (5 mg/kg, cat #: BE0146, Bio X cell, NH, USA) or anti-mouse PD-L1 (B7-H1) antibody (cat #: BE0101, Bio X cell, NH, USA) was administered on Days 6, 8, 11, 14, 17, and 20. (4). CIS (MedChemExpress, Monmouth Junction, NJ, USA, 3 mg/kg, dissolved with 0.5% sodium

carboxymethyl cellulose (CMC-Na, MedChemExpress, Monmouth Junction, NJ, USA) according to manufacturer's instructions) was administered twice a week for a total of five times[55]. (5). Combined treatment was administered with two or three of the above drugs. (6). Equal amounts of isotype control antibody, 0.5% CMC-Na, or PBS were used. Bodyweight of the mice were measured every 2 days from Day 6 with electronic scales. Subcutaneous tumor size was measured every

**Fig. 5 | Depletion of NK cells reverses the antitumor effects of CD39i in vivo. A–C** CD39i significantly inhibited the tumor growth and improved the overall survival rate in normal mice, but the effect of CD39i was remarkably reversed in the NK cell-depleted mice (*n* = 10 for each group). Comparisons of tumor growth curves were performed by a two-way ANOVA test followed by Tukey's multiple comparison test. The log-rank (Mantel–Cox) test was used to compare the survival curves. **D** CD39i treatment increased the proportion of tumor infiltrated cDC1 (CD45 + CD11c + MHC II + CD103 + CD11b-) in control mice, but not in NK cells absent mice. The flow cytometry analyses were repeated 3 times with 5 samples in

each group. **E–G** CD39i was confirmed to inhibit the progression of BC and improve the prognosis of mice, but it failed to effectively inhibit the growth of subcutaneous tumors or improve the mouse prognosis in cDC1-deficient *Batf3*[−/−] mice, *n* = 10 for each group. Comparisons of tumor growth curves were performed by a two-way ANOVA test followed by Tukey's multiple comparison test. The log-rank (Mantel–Cox) test was used to compare the survival curves. Source data are provided as a Source Data file (**A**, **C**, **E**, **G**). Bar graphs show the mean ± SEM, and *P*-values <0.05 were considered significant: **P* < 0.05; ***P* < 0.01; ****P* < 0.001; *****P* < 0.0001.

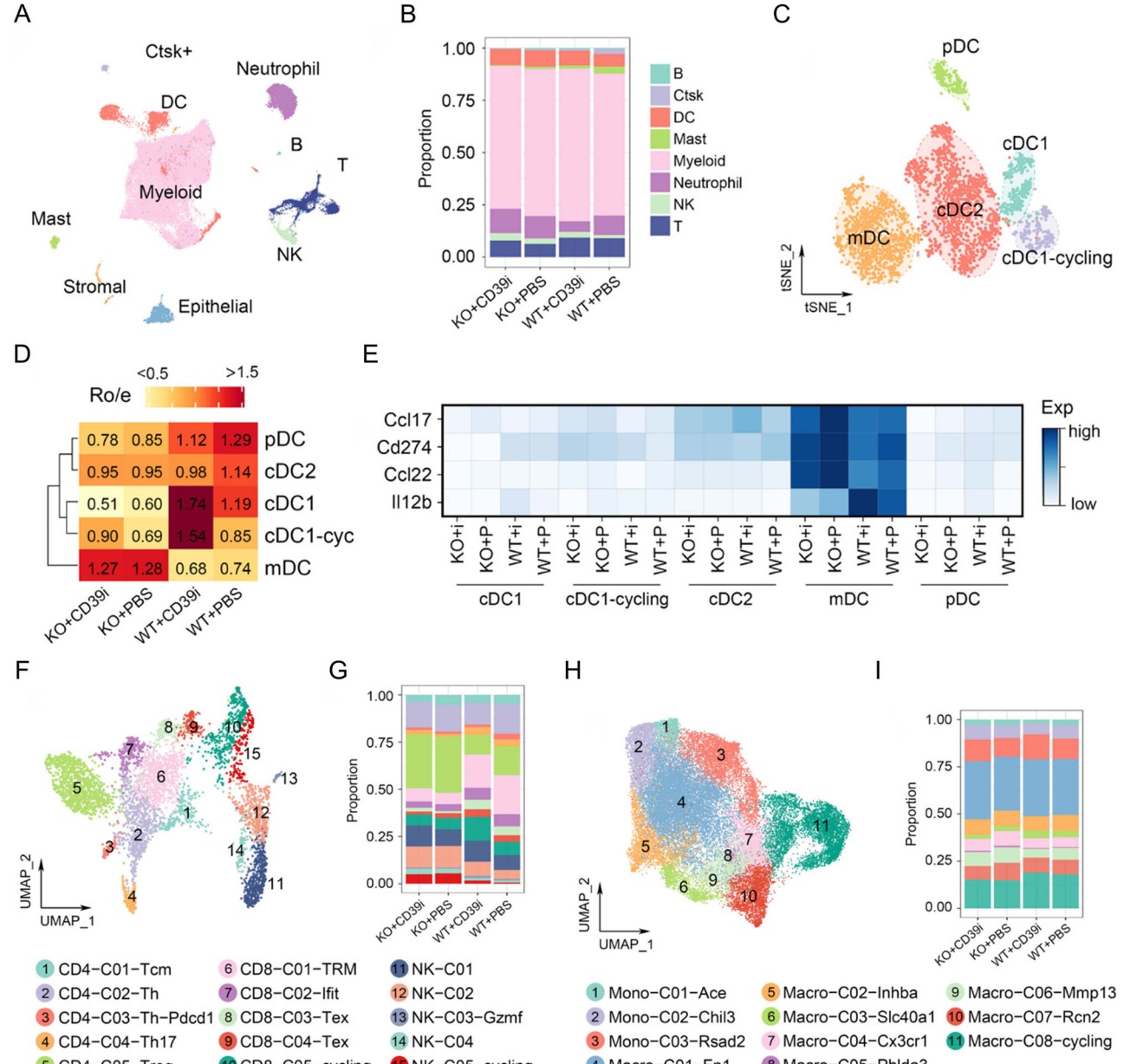

**Fig. 6 | The effects of *Batf3*[−/−] cDC1-deficient on tumor microenvironment and CD39i efficacy. A** Single cells from WT and *Batf3*[−/−] cDC1-deficient mice treated with or without CD39i were clustered into 10 major clusters, 3 subcutaneous tumors from 3 mice in each group mixed 1:1:1. **B** The proportions of 8 major clusters without stromal cells and epithelial cells. **C** The DCs were clustered into 5 clusters based on classical markers. **D** Ratios of observed to expected cell numbers (Ro/e) for each DC subcluster in different groups, Ro/e > 1.1 was considered as enriched in

the group. **E** Compared to WT mice, mDC from *Batf3*[−/−] mice expressed more Ccl17 and Ccl22 and less Il12b. **F** Lymphocytes were clustered into 15 clusters. **G** CD39i treatment significantly increased the proportions of proliferating CD8 + T cells and NK cells, but decreased the proportion of Treg and Th cells in *Batf3*[+/+] mice. In *Batf3*[−/−] mice, only a slight increase in NK cells was observed. **H, I** CD39i was not responsive to mononuclear macrophages in both *Batf3*[+/+] mice and *Batf3*[−/−] mice.

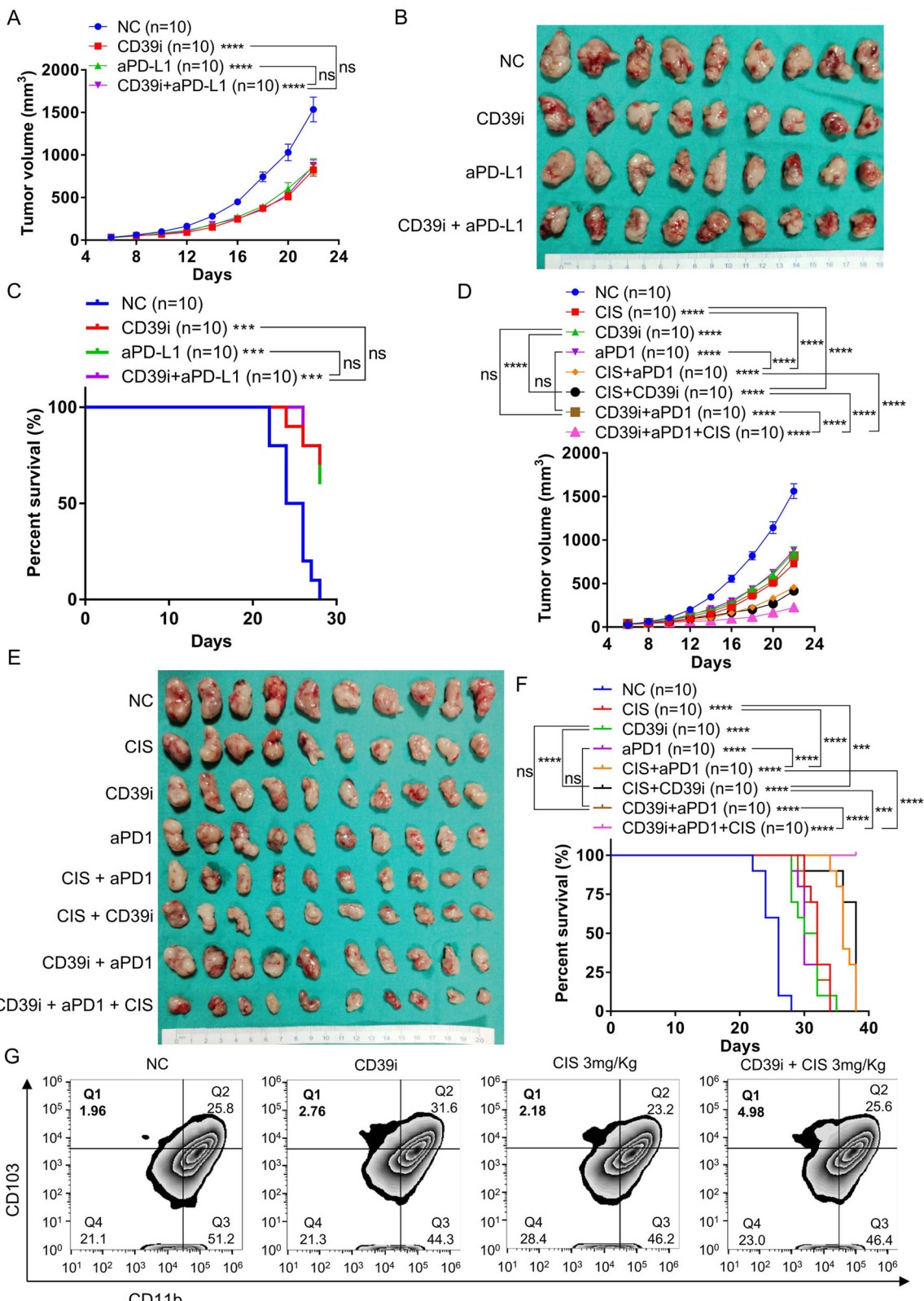

2 days from Day 6 with digital callipers. The tumor volume was calculated by the formula: tumor volume (mm³) = 0.5 × longest dimension × perpendicular dimension[2]. For single-cell RNA-Seq and flow cytometry analyses of the subcutaneous tumors, the mice were sacrificed on Day 18 and Day 22, respectively. For the survival experiments, the mice with bodyweight loss exceeding 20%, that became moribund, or whose subcutaneous tumor volumes reached 2000 mm³ or became

necrotic were considered dead[56–59]. The maximal permitted tumor size/burden was 2000 mm³, and when the tumor volume reached 2000 mm³, the mice were euthanized as the humanitarian end-point of the study. During this study, some tumors reached 1800–1900 mm³ at the penultimate measurement point without an end-point event (>2000 mm³). Because of the accelerated growth of MB49 cells subcutaneous tumors, some tumors reached more than 2000 mm³ in last

**Fig. 7 | Effects of CD39i combined with commonly used anti-bladder cancer drugs. A–C** Monotherapy with CD39i or aPD-L1 antibody conspicuously inhibited the tumor growth and improved the mouse prognosis, but the combination therapy with CD39i and aPD-L1 did not show any synergistic effect, $n = 10$ for each group. **D–F** The combination strategies, CD39i + CIS and aPD1 + CIS, but not CD39i + aPD1, had a significant synergistic effects on inhibiting tumor growth and improving prognosis, and the combination of CD39i, CIS, and aPD1 had the strongest antitumor efficacy in vivo, $n = 10$ for each group. Comparisons of tumor growth curves were performed by a two-way ANOVA test followed by Tukey's

multiple comparison test. The log-rank (Mantel–Cox) test was used to compare the survival curves. **G** CD39i (but not 3 mg/kg cisplatin alone) treatment significantly increased the proportion of tumor infiltrated cDC1 cells, and cisplatin at dose of 3 mg/kg had a synergistic effect with CD39i on increasing the proportion of cDC1 in tumors. cDC1: CD45 + CD11c + MHC II + CD103 + CD11b−. The flow cytometry analyses were repeated at 3 times with 5 samples in each group. Source data are provided as a Source Data file (A, C, D, F). Bar graphs show the mean ± SEM, and $P$-values <0.05 were considered significant: *$P < 0.05$; **$P < 0.01$; ***$P < 0.001$; ****$P < 0.0001$.

2 days (in extreme cases, the size of the tumor increased by up to 800 mm³ in last 2 days). For these mice, we stopped the experiment immediately after measuring the tumor volume. The tumor volume measurements are provided in the Source Data file.

## Cell isolation

Cell isolation was performed according to a previous procedure[60–62] with some adaptations. Briefly, after 4 mice were sacrificed, subcutaneous tumors were collected and cut into small pieces (<2 mm in diameter) and then digested with 10 ml of dissociation solution (RPMI 1640 medium containing 10% FBS, 1 mg/mL collagenase type IV (Biosharp, Hefei, China), 100 μg/mL DNase I (Biosharp, Hefei, China), and 2.5 μg/mL hyaluronidase (Biosharp, Hefei, China)) for 60 min on a 37 °C shaker. Next, 4 ml RPMI 1640 medium containing 10% FBS was added to dilute the suspensions. Then, the cell suspensions were passed through 70 μm cell strainers, and the lower layer was collected after centrifuging at $400 \times g$ for 10 min. Subsequently, the erythrocytes were lysed with red blood cell lysis buffer (Biosharp, Hefei, China) according to the manufacturer's instructions and then washed and resuspended the remaining single cells in PBS.

## Flow cytometry assay

Flow cytometry assays were performed as described previously[62]. In brief, the single cells resuspended in PBS were first stained with antibodies specific for cell surface markers and with Fixable Viability Stain 780 (1: 100, APC-Cyanine7, cat #: 565388, BD HorizonTM), and then washed with PBS for 2 times before staining with antibodies specific for intracellular markers according to the manufacturer's instructions. The single-cell suspensions were stained with the following antibodies purchased from BD (San Diego, USA): anti-mouse CD45-FITC (1: 100, cat #: 553079), anti-mouse CD4-BB700 (1: 100, cat #: 566407), anti-mouse CD8a-PE-Cyanine7 (1: 100, cat #: 552877), anti-mouse CD3e-BV510 (1: 50, cat #: 563024), anti-mouse CD335/NKp46-Alexa 647 (1: 100, cat #:560755), anti-mouse IFN-γ-BV421 (1: 50, cat #: 563376), anti-mouse Ki-67-BV605 (1: 50, cat #: 567122), anti-mouse TCF-7/TCF-1-Alexa 647 (1: 50, cat #: 566693), anti-mouse CD279 (PD-1)-APC-R700 (1: 50, cat #: 565815), anti-mouse CD11b-PE-Cyanine7 (1: 50, cat #: 552850), anti-mouse I-A/I-E (MHC-II)-BB700 (1: 50, cat #: 746086), anti-mouse CD11c-BV605 (1: 50, cat #: 563057), and anti-mouse CD172a/SIRPα-PE (1: 50, cat #: 560107). The following antibodies purchased from Bio-Legend (San Diego, USA): anti-mouse Perforin-PE (1: 50, cat #: 154306), anti-mouse Granzyme B Recombinant-PE/Dazzle™ 594 (1: 50, cat #: 372216), anti-mouse CD27-PE (1: 50, cat #: 124210), anti-mouse CD103-APC (1: 50, cat #: 121413), and anti-mouse/rat XCR1-BV421 (1: 50, cat #: 148216). Flow cytometry was performed using a flow cytometer (BD LSRFortessa X-20, USA), and analysis was performed using FlowJo: Flow Cytometry Analysis Software, v10.0.7 (Tree Star). All the flow cytometry analyses were repeated 3 times with 5 samples in each group.

## Magnetic-activated cell sorting (MACS) separation of CD45 + cells

CD45 + cells were isolated from subcutaneous tumors of different groups using CD45 immunomagnetic beads. Briefly, single cells obtained by dissociation of subcutaneous tumor tissues were

resuspended in MACS buffer, and then total cells were incubated with FcR blocking buffer and CD45 microbeads for 30 min at 4 °C. Subsequently, the cells were washed with MACS buffer and centrifuged at $400 \times g$ for 10 min. After removing the supernatant, the cell pellets were resuspended in MACS buffer and the LS magnetic columns were used to obtain the CD45 + cells according to the manufacture's protocol. All reagents and LS magnetic columns used were from Miltenyi Biotec Inc., Bergisch Gladbach, Germany.

## Single-cell RNA-sequencing

Single-cell RNA-sequencing was performed as described in our previous work[14]. In brief, subcutaneous tumors from 3 mice in each group were dissociated to obtain single-cell suspensions of immune cells, which were then mixed 1:1:1 and were loaded on a 10X Chromium Controller (10X Genomics) to generate nanoliter-scale gel beads-in-emulsions (GEMs). Barcoded scRNA-seq libraries were prepared using the Chromium Single Cell 3' v3 Reagent Kit. Then, the GEMs were used to generate barcoded, full-length cDNA through reverse transcription reactions. Next, the barcoded, full-length cDNA was used for library construction via fragmentation, end repair, A-tailing, ligation to an index adaptor, and amplification by PCR. The final libraries were sequenced on a HiSeq X Ten platform (Illumina), and 150 bp paired-end reads were generated[14].

## Raw data processing, quality control, and downstream analysis

Cell Ranger (version 5.0.0) was used to process the raw data and generate the UMI matrix. Cells with fewer than 1000 UMIs, or with over 20% percent of transcripts derived from mitochondria were considered as low-quality cells and were discarded. All the downstream analyses were performed with Seurat (v 3.0.1) in the R environment (version 3.6.1). Considering that a library of each sample was constructed respectively, sample IDs were used to remove potential batch effects with Harmony (https://github.com/immunogenomics/harmony)[63]. After principal component analysis (PCA) analysis, the top 50 PCs were used to perform tSNE or UMAP analysis. Clusters found by the FindClusters function of Seurat were annotated with known markers as listed in the Results. Heatmaps of selected genes were visualized with scanpy (version 1.7.1)[64].

## Cell–cell communication analysis with CellPhoneDB 2[29,30]

In order to construct the communication network, CellPhoneDB 2[29,30] analysis was performed to evaluate ligand-receptor pairs between immune cell clusters. A normalized expression matrix was used as the input data. Only genes expressed in >10% single cells in at least one cell subgroup were considered in this analysis.

## Group distribution of DC subclusters

The ratio of observed to expected cell numbers (Ro/e) for each DC subcluster in different groups was calculated to quantify the group preference of each cluster, as previously described[65,66]. Ro/e > 1.1 was considered as enriched in the group.

## Human samples

Formalin fixed, paraffin-embedded human tissue arrays (HBlaU079Su01) were purchased from Shanghai Outdo Biotech Co.,

Ltd. (China), including a total of 63 cancer tissues and 16 cancer-adjacent normal tissues from patients with bladder cancer. These tissue arrays were subjected to immunohistochemistry (IHC) and immunofluorescence (IFC) staining assays, as described below. The detailed clinical information was downloaded from the company's website.

### IHC staining assay

The 4 µm sections cut from an array block of BC patients and sub-cutaneous tumors of WT and Batf3 KO mice were deparaffinized with heat at 60 °C for 30 min followed by three 10-minute washes with xylene. Then, the paraffin sections were rehydrated by washing for 5 min in absolute ethanol I, absolute ethanol II, 85% alcohol, 75% alcohol and distilled water in sequence. A microwave oven was used to retrieve the antigen by heating the samples in citric acid antigen retrieval buffer (pH 6.0). Subsequently, the section was incubated in 3% hydrogen peroxide for 30 min to block the endogenous peroxidase activity, and the tissues were sealed for 30 min at room temperature with 3% BSA added to block nonspecific binding. Next, the tissue sections were incubated with anti-CD39 (1:1000, Abcam, ab223842) or anti-BATF3 (1:100, Abbkine, ABP57435) primary antibodies overnight in a wet box at 4 °C. Then, the HRP anti-Rabbit IgG antibody (1: 200, Servicebio, G1213) and DAB color developing solution were used for immunodetection at room temperature. Finally, the prepared sections were scanned as high-resolution digital images at 5.4× using a Pan-noramic MIDI II scanner (3DHISTECH Ltd., Budapest, Hungary).

The expression level of CD39 staining was assessed using Remmele and Stegner's semiquantitative immunoreactive score (IRS) scale[67–69]. Briefly, the CD39 signal was mainly detected in the plasma membrane and extracellular space. The staining intensity for CD39 was scored on a scale of 0–3 (0, negative; 1, weak; 2, moderate; and 3, strong). The CD39 positive cell proportion was also scored as follows: 0, no staining; 1, 1–25%; 2, 26–50%; 3, 51–75%; and 4, 76–100% stained cells. Then, the staining intensity and positive cell proportion scores were multiplied to obtain a final score.

### IFC staining assay

Tissue array slicing, dewaxing, rehydrating, antigen retrieval, and nonspecific binding blocking steps were performed as described above in the IHC staining assay. Next, the tissue sections were incubated with primary antibodies against CD8 (1:100, Abcam, ab178089) and LAG3 (1:1000, Abcam, ab209236) overnight at 4 °C, and then incubated with the corresponding secondary antibodies at room temperature for 50 min in the dark. Subsequently, the nucleus was stained with DAPI solution at room temperature for 10 min followed by intrinsic fluorescence quenching. Finally, the prepared section was scanned as high-resolution digital images using a Pannoramic MIDI II scanner (3DHISTECH Ltd., Budapest, Hungary). DAPI glows blue by UV excitation, CD8 was labeled with red fluorescence, and LAG3 was labeled with green fluorescence. The IRS of CD8 was calculated by multiplying the mean number of CD8 positive cells per high magnifi-cation by 0.6. The IRS of CD8 and LAG-3 double positive cells was calculated by multiplying the ratio of CD8 + LAG-3 + /CD8 + cells by 12.

### ELISA

The quantitative detection kit for mouse XCL1 was purchased from HYCEZMBIO (cat #: HY021623M, Wuhan, China), and used to detect the tumor XCL1 protein level of mice after treatment with isotype control antibody, CD39i, aNK1.1, and CD39i + aNK1.1, respectively. The operation process was carried out according to the product instruc-tions provided by the company.

### Statistical analysis

The two-side unpaired Student's $t$-test was used for two-group com-parisons of values. Comparisons of tumor growth curves were performed by a two-way ANOVA test followed by Tukey's multiple comparison test. The log-rank (Mantel–Cox) test was used to compare the survival curves. A linear regression model (Pearson's correlation) was used to determine the individual correlations between different variables. All analyses were finished using the GraphPad Prism 6.02 software (GraphPad Software Inc., San Diego, CA, USA). All the error bars represent the standard error of the mean (SEM). $P$-values < 0.05 were considered significant: *$P$ < 0.05; **$P$ < 0.01; ***$P$ < 0.001; ****$P$ < 0.0001. Unless otherwise indicated, the asterisks indicate the statistical comparison to the control group.

### Reporting summary

Further information on research design is available in the Nature Portfolio Reporting Summary linked to this article.

## Data availability

The public TCGA-BLCA data used in this study are publically available in the UCSC XENA database under accession code http://xena.ucsc.edu/. The single-cell RNA-sequencing publicly available data of the 8 bladder cancer and 3 paracancer tissues of bladder cancer patients used in this study are available in the GSA-Human database under the accession code HRA000212 and in the SRA datasets under BioProject PRJNA662018[14]. The dataset in GSA is available under restricted access, while the dataset in SRA is not. The raw single-cell RNA-sequencing data of mice in different treatment groups (untreated, CD39i, WT-untreated, WT-CD39i, *Batf3*$^{-/-}$-untreated, and *Batf3*$^{-/-}$-CD39i) gener-ated in this study are deposited in the Gene Expression Omnibus (GEO) database under accession number GSE189127. The IMvigor210 dataset[21] was exported from the R package IMvigor210CoreBiologies under accession code http://research-pub.gene.com/IMvigor210CoreBiologies/. Source data as provided with this paper in the Source Data file. The remaining data are available within the Article, Supplementary Information or Source Data file. Source data are provided with this paper.

## Code availability

The computer codes used in this study are deposited in github under accession code https://github.com/AndersonHu85/CD39_code.git.

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

## Acknowledgements

This work was supported by grants from the National Natural Sciences Foundation of China (82072852, K.C.) and the Fundamental Research Funds for the Central University, HUST (2019kfyRCPY004, K.C.; 2021GCRC040, K.C.). The diagrams of the cell–cell communication network were generated in BioRender (https://biorender.com/). The authors thank Shanghai OE Biotech Co., Ltd. (China) for providing single-cell RNA-seq and Dr. Yongbing Ba and Yao Lu for assistance with bioinformatics analysis. The authors thank Shanghai Universal Biotech Co., Ltd. (China) for helping to buy antibodies (Bio X cell, NH, USA; BD, San Diego, USA) and thank Linglong Zhang for the guidance on flow cytometry.

## Author contributions

K.C., J.H., and Z.C. conceived the idea for the study. L.L., Y.H., C.D., and J.H. designed and performed the experiments. J.H., L.L., and Y.H. analyzed the data. L.L. and Y.H. wrote the manuscript. K.C., J.H., Z.C., and Z.T. helped revise the manuscript. All authors reviewed and approved the manuscript.

## Competing interests

The authors declare no competing interests.
