## [Peer Review File · Nature Communications]

Single Cell Sequencing Reveals that CD39 Inhibition Mediates Changes to the Tumor Microenvironment in Bladder CancerReviewers' Comments:

Reviewer #1:

Remarks to the Author:

The manuscript by Liu, Hou et al. identifies CD39 as a highly expressed marker in bladder cancer (BC) tissues. Using mouse models and single-cell transcriptomics, the authors show that inhibition of CD39 inhibits BC tumour growth, and that this effect seems to be mediated by cDC1s. The authors propose CD39 a potential therapeutic target and show that in a mouse tumour model, a combined regimen of CD39inhibition and Cisplatin plus PD-1 blockade leads to better outcome than Cis + PD-1 blockade alone.

The work has many merits. The manuscript is coherent and tells a compelling story, and the results are potentially of high relevance. In my opinion, the conclusions are well supported by the data presented. The experimental approaches are well chosen, and the results are for the most part logically and clearly presented.

In my view, the shortcomings of the manuscripts are quite minor, and they are mostly related to language and presentation, and in some parts to lack of detail. I would suggest that the following minor issues are addressed before publication:

Lines 80-85 and Figure 1A: Based on the text, it is not clear how this data was produced and processed. This should be described in more detail.

Figures 1A, 2I and 5K: the expression of the known cell type-specific transcripts should be presented (e.g. as supplementary figures)

Figure 1E: X-axis unit is missing

Language could benefit from some revising, e.g. the sentence on lines 144-145 is not very easy to understand upon first reading.

In addition, there are several spelling errors, such as:

Fig1 A myloid – myeloid

Line 19 words "inhibition of" missing?

Line 20 reprogram -> reprogramming ?

Line 50 interquartil -> interquartile

Line 60 word "is" missing?

Line 109: mucusa -> mucosa

Line 157: significantly -> significant

Reviewer #2:

Remarks to the Author:

This manuscript describes CD39 as overexpressed at the single cell level in myeloid and other stromal cells from scRNAseq data derived from human bladder tumors and highlights CD39 expression correlates with poor outcomes in the TCGA bladder cancer cohort. The authors go on to dissect the impact of CD39 inhibition on tumor growth in an MB49 SC model system along with the impact of CD39 inhibition (with or without other therapies) on the composition of the tumor microenvironment at single cell resolution.

1. Though the intro suggests that the single cell cohort was previously described in the authors prior publication, it is very difficult to follow in the current manuscript the human cohort utilized, how many samples, and other details of the sequencing. At least some detail is needed in the current study. It is even difficult in places to understand what data was from human versus mouse.

2. How was CD39 identified as important and the focus of the manuscript? This is not clear from the intro or the results and really forms the basis for the manuscript.

3. POM-1 is introduced in the introduction with really no further details and then the term is used again in the results. What is POM-1?

Reviewer #3:

Remarks to the Author:

The current manuscript by Liu and colleagues is designed to investigate the mechanism of action for therapies designed to target CD39 in the context of cancer treatment. CD39 catalyses the conversion of immune activating ATP into AMP, which is subsequently converted to adenosine by the actions of CD73. This is of interest to the field as several therapeutics have been designed to target CD39, and more broadly, CD73 and the downstream adenosine receptors and yet the mechanism of action for these inhibitors remain broadly undefined. The manuscript principally investigates this through employing scRNA-Seq, which identifies an increase in cDC1 population that is linked to therapeutic effect. Overall however, quite limited validation of these observations are made and the link between these findings and the combination therapy of CD39i and cisplatin is not clear. There are therefore several additional experiments that should be considered to further strengthen the conclusions of the work.

Major points

- The authors postulate that CD39 inhibition enhances NK cell activity leading to increased recruitment of cDC1s. However, some key experiments required to validate this mechanism are absent. For example confirming that CD39 inhibition fails to enhance cDC1 numbers in the absence of NK cells. NK deficient mice or NK depleting antibodies could be used to confirm this. Do NK cells enhance their expression of XCL1 in the context of CD39 inhibition?

Figure 4- The NK1.1 staining is not convincing and thus undermines the message that CD39i enhances NK cell infiltrate. The authors should utilise alternative methods of identifying NK cells such as Nkp46 to try and improve this. It may also be of interest to concurrently perform a basic analysis of NK cell phenotype too e.g. CD11b, CD27, Granzyme B.

- Does CD39 inhibition enhance cDC1 function? Some experiments to show improved T cell priming would be useful to validate this mechanism.

-What effect does cisplatin have on the immune effects mediated by CD39 inhibition? This should be explored through flow cytometry or RNA analyses. Is there interplay with the cDC1 population for example? Without this detail the rationale to combine these treatment types is not clear.

- The authors have an opportunity to investigate why anti-PD-L1 does not synergise with CD39i in this model. PD-1 blockade is thought to enhance the activation/ expansion of precursor exhausted T cells. This cell population is not annotated in the single cell dataset but can the authors detect changes in this population by scRNA-Seq or FACS following CD39i?

Minor comments-

- The conclusions are based entirely upon the CD39 inhibitor POM-1. Whilst POM-1 has reasonable selectivity for CD39, other enzymes are known to be inhibited by this drug and so ideally results would be confirmed using a genetic-based approach. In the absence of this data, I recommend authors simply discuss this potential caveat.

- It was not apparent how frequently flow cytometry analyses were performed e.g. Figures 2D-G as I could not see this information in the Figure legend. As it stands it could be interpreted that this experiment was performed only once, which would be unacceptable.

- Figure 3J shows the clustering associated with the distinct myeloid sub populations. But this graph is not annotated to indicate genes characteristic of each cluster and so adds little value to the manuscript in its current form.

- Line 175 the authors conclude that CD39i promotes expansion of CTL cells but proliferation is not directly assessed and so this data could reflect increased survival. The language here should be modified accordingly.

Figure 4A- The cell populations on the scRNA-Seq plot do not appear to match the labels below?

-Line 186- The authors suggest XCl1 release from NK cells may activate cDC1...one assumes this should state "recruit cDC1?"

Reviewer #4:

Remarks to the Author:

This study by Liu et al., uses single cell sequencing (scRNA-seq), human samples and mouse tumor models to study the role of CD39 (an "ecto-enzyme") in bladder cancer immunity and progression. The authors use scRNA-seq plots to show restricted expression of CD39 to immune cells (myeloid, lymphoid), stromal and endothelial populations. With this, studies are focused on assessing the importance of CD39 in T cell function and activity. TCGA tissue microarrays are used to assess expression changes of CD39 in tumor cores as compared to normal tissues. Using the mouse MB49 bladder cancer model, the investigators conduct a series of S.Q. tumor treatment studies using the POM-1 inhibitor to assess whether CD39 regulates T cell proliferation and expression changes of inhibitory checkpoint proteins. POM-1 treatment studies of S.Q. MB49 tumors are also used to monitor tumor progression and mouse survival. Using a genetic approach, the authors propagate MB49 S.Q. tumors in mice deficient for Batf3 (Batf3 KO) - a transcription factor with important functions in the development of CD8+ DCs - to suggest that CD39 inhibition is mediated through DC populations. Finally, the authors use the MB49 tumor model to suggest that CD39 inhibition is not synergistic with immune checkpoint inhibition but is synergistic with cisplatin chemotherapy.

Despite these experiments, this paper was difficult to read and is not suitable for publication for the following Major Issues:

(1) Previous studies have shown CD39 is increased in human bladder cancer including expression in immune cells and scRNA-seq data (PMID: 34381563, 33790578).

(2) Much of the presented data is not convincing either due to lack of important raw data or experimental controls. All this information could all be included as supplemental information.

(3) Animal studies use cells implanted S.Q. rather than into the host bladder making extrapolation to human disease difficult. Well established methods for implanting cells to the bladder wall exist.

(4) The entire paper was difficult to read including extensive grammar, syntax, and other writing issues. Even the figures are loaded with spelling mistakes (e.g., Fig 2B, vertical axis, "Volumn")

Specific Issues:

Figure 1

A, how many replicates were completed for these analyses? Please show independent TSNE replicate plots

C, the IHC images are too small to distinguish the cited lineage specific expression of CD39. Pls include low and high magnifications. Please include a low mag scan the CD39 stained TMA in supplemental as well as independent cores each from normal and cancer. At least 5 cores from normal versus tumor.

D, how was this scoring completed? What do values of 5, 10 and 15 represent?

G, this data is not clear as there are a different number of events shown for normal and tumor. Cell numbers should be the same for both TSNE plots.

K, IF staining is not clear especially LAG3 expression. Pls include panels of single channel marker expression, low-high mag views of each stain, and more fields from tumors.

Figure 2

A,H the cartoons are not helpful and should be removed (lack units, incomplete, redundant)

B, please show gross images PBS and POM-1 that match the indicated error bars

C, how is survival defined

D, please include FACS plots for at least 5 ctrl and 5 POM1 tumors with equal events analyzed

I, clusters are not labelled

J, include bar graphs for all ctrl and POM-1 replicates cited in E-G

Figure 4

A-B, It is unclear how this shows that T cells are cycling more. Please include individual tumors plots to support this claim

A-B, Suggest using an alternative assay (flow based) to support the claim that CD39 inhibition alters T cell activity.

E, FACS plots show different numbers of events. Pls keep standardize and include at least 5 replicate plots in the supplemental.

H-I, this is completely confused and poorly explained.

Figure 5.

A, cartoon is not clear and redundant. Suggest removing

B, tumor growth curves are not discernable as presented. Either fix the vertical scale or format. Where are the error bars?

C, please show the absence of Batf3 by IHV to confirm that the expression difference between WT and KO mice using implanted tumors. Show low and high mag images of the MB tumors implanted to WT and KO mice.

B-C, the IACUC limit for individual SQ tumors in mice is 2000 mm³, thus is unclear how the authors have produced tumors (individual data points) of tumors exceeding 4000mm³ and up to 6000mm³. Please show gross images of these SQ tumors and how these related to the error bars shown.

Reviewer #1:

We thank the reviewer for the thoughtful comments, and agree with the reviewer that the “The conclusions are well supported by the data presented”. The issues in language and presentation, as well as the lack of detail in some parts raised by the reviewer, have been properly addressed below.

1. Lines 80-85 and Figure 1A: Based on the text, it is not clear how this data was produced and processed. This should be described in more detail.

The reviewer requested “Figure 1A should be described in more detail”. We agree with the reviewer and according to their advice. More description of **Figure 1A** data was shown in the Results of the manuscript “After quality control and batch effect removal, the single cells from 8 BC samples and 3 para tumor samples¹⁴ were clustered into 10 major clusters (**Figure 1A & SFigure 1C**), and the raw data is available in GSA-Human under the accession code HRA000212 and in SRA datasets under BioProject PRJNA662018. Known markers were used to annotate these cell types (**SFigure 2A**).”

2. Figures 1A, 2I and 5K: the expression of the known cell type-specific transcripts should be presented (e.g. as supplementary figures)

The reviewer’s point is well taken. The expression of the known cell type-specific transcripts of Figures 1A, 2I and 5K were presented in **SFigure 2A, SFigure 5A, and SFigure 9B, C**.

3. Figure 1E: X-axis unit is missing

We thank the reviewer for pointing out this issue, and we have re-organized **Figure 1E**.

4. Language could benefit from some revising, e.g. the sentence on lines 144-145 is not very easy to understand upon first reading.

In addition, there are several spelling errors, such as:

Fig1 A myloid – myeloid

Line 19 words “inhibition of” missing?

Line 20 reprogram -> reprogramming ?

Line 50 interquartil -> interquartile

Line 60 word “is” missing?

Line 109: mucusa -> mucosa

Line 157: significantly -> significant

Thanks to the reviewer for putting forward these revision suggestions, which are very helpful for us to further improve the manuscript. According to the reviewer's suggestion, we rearranged the sentences in lines 144-145 as “However, although the proportion of DC subgroups have been relatively remodelled, we have not observed any significant changes of functional genes in each cell subgroup after CD39i treatment.” and corrected all spelling mistakes put forward by the reviewer.

Reviewer #2:

Thanks for the reviewer's comments. The issues raised by the reviewer have been addressed below.

1. Though the intro suggests that the single cell cohort was previously described in the authors prior publication, it is very difficult to follow in the current manuscript the human cohort utilized, how many samples, and other details of the sequencing. At least some detail is needed in the current study. It is even difficult in places to understand what data was from human versus mouse.

The reviewers expressed their concern “It is very difficult to follow in the current manuscript the human cohort utilized, how many samples, and other details of the sequencing” and asked “At least some detail is needed in the current study”. We agree with the reviewer their advice.

Firstly, we have added the details of the sample in the Data availability section as follows.

“The single-cell RNA sequencing data of 8 bladder cancer and 3 para-cancer tissues of bladder cancer patients used in this paper is available in GSA-Human under the accession code HRA000212 and in SRA datasets under BioProject PRJNA662018. TCGA-BLCA datasets were downloaded from UCSC XENA (<http://xena.ucsc.edu/>). In addition, the IMvigor210 dataset²⁰ was exported from the R package IMvigor210CoreBiologies.

All the single-cell RNA sequencing data of mice in different treatment groups have been deposited in the Gene Expression Omnibus (GEO) database (<https://www.ncbi.nlm.nih.gov/geo/>) under accession number GSE189127 by password: afahcusgdnixhef. All other relevant raw data and computer code are available in the article, supplementary information, or from corresponding author on reasonable request.”

Secondly, we have further described our previous work in the Introduction section as “We have generated a single-cell atlas of 8 tumor samples and 3 para tumor samples from BC patients, revealed distinct component and microenvironment features between different molecular subtypes, which may lead to differential response to chemotherapy or other targeted therapy, reminding the necessity of development of personalized therapy¹⁴.”

Thirdly, we have described the raw data processing, quality control and downstream analysis of single-cell RNA sequencing data in the Materials and Methods section as “Cell Ranger (version 5.0.0) was used to process the raw data and generate the UMI matrix. Cells with fewer than 1000 UMIs, or with over 20% percent of transcripts derived from mitochondrial were considered as low-quality cells and were discarded. All the downstream analyses were done with Seurat (v 3.0.1) in R environment (version 3.6.1). Considering that the library of each sample was constructed respectively, sample IDs were used to remove potential batch effect with Harmony⁶³. After principal component analysis (PCA) analysis, top 50 PCs were used to perform tSNE or UMAP analysis. Clusters found by the FindClusters function of Seurat were annotated with known markers as listed in the Results. Heatmaps of selected genes were visualized with scanpy (version 1.7.1)⁶⁴.” We thank the reviewers for their comments and suggestions, which have further strengthened the manuscript.

14. Chen Z, et al. Single-cell RNA sequencing highlights the role of inflammatory cancer-associated fibroblasts in bladder urothelial carcinoma. *Nature communications* 11, 5077 (2020).
20. Mariathasan S, et al. TGF β attenuates tumour response to PD-L1 blockade by contributing to exclusion of T cells. *Nature* 554, 544-548 (2018).
63. Korsunsky I, et al. Fast, sensitive and accurate integration of single-cell data with Harmony. *Nat Methods* 16, 1289-1296 (2019).
64. Wolf FA, Angerer P, Theis FJ. SCANPY: large-scale single-cell gene expression data analysis. *Genome biology* 19, 15 (2018).

2. How was CD39 identified as important and the focus of the manuscript? This is not clear from the intro or the results and really forms the basis for the manuscript.

The reviewer's point is well taken. By analyzing our previously published data (8 tumor samples and 3 para tumor samples from BC patients)¹⁴, we found that numerous immune checkpoints are significantly increased in tumor-derived lymphocytes (**SFigure 1A**). We then analyzed TCGA-BLCA data using GEPIA¹⁷, an online tool available at <http://gepia.cancer-pku.cn/>, and found that only CD39 was significantly associated with BC progression among these immune checkpoints (**SFigure 1B**). Through single cell RNA sequencing analysis of bladder cancer tissue samples, we found that tumor infiltrated CD8⁺ T cells and NK cells showed a significantly higher level of CD39, along with increased LAG3, another marker of T cell exhaustion (**Figure 1F-G**). In the TCGA-BLCA dataset, the CD39 expression level was positively correlated with T cell exhaustion signature (**Figure 1H**). Furthermore, by using a tissue array containing BC and para-cancer tissues, we found that the expression level of CD39 was positively correlated with the proportion of exhausted CD8⁺ T cells, but was not related to the abundance of CD8⁺ T cells (**Figure 1K-M & SFigure 3B**), indicating that CD39 is related to T cell dysfunction degree, while not associated with immune cells infiltration level of BC. Based on these findings, we hypothesized that CD39 might play an important role in the regulation of tumor microenvironment, then a series of experiments were conducted to explore the function and underlying mechanism of CD39 blocking on immune microenvironment of bladder cancer. We have re-organized the Figures and additional experimental results have been added in the Result section. We appreciate the comments and suggestions of the reviewer, which have further strengthened the manuscript.

14. Chen Z, et al. Single-cell RNA sequencing highlights the role of inflammatory cancer-associated fibroblasts in bladder urothelial carcinoma. *Nature communications* 11, 5077 (2020).
17. Tang Z, Li C, Kang B, Gao G, Li C, Zhang Z. GEPIA: a web server for cancer and normal gene expression profiling and interactive analyses. *Nucleic acids research* 45, W98-w102 (2017).

3. POM-1 is introduced in the introduction with really no further details and then the term is used again in the results. What is POM-1?

Thank the reviewers for pointing out this issue. We have given a detailed explanation of POM-1 as “With a subcutaneous model, we demonstrated that inhibition of CD39 (CD39i) by sodium polyoxotungstate (POM-1, a novel ENTPDase inhibitor)¹⁶ could inhibit BC growth and improve prognosis in vivo” in the Introduction section.

16. Wall MJ, Wigmore G, Lopatár J, Frenguelli BG, Dale N. The novel NTPDase inhibitor sodium polyoxotungstate (POM-1) inhibits ATP breakdown but also blocks central synaptic transmission, an action independent of NTPDase inhibition. *Neuropharmacology* 55, 1251-1258 (2008).

Reviewer #3:

We thank the reviewer for the thoughtful comments that the “This is of interest to the field as several therapeutics have been designed to target CD39, and more broadly, CD73 and the downstream adenosine receptors and yet the mechanism of action for these inhibitors remain broadly undefined.” And agree with the reviewer that “There are therefore several additional experiments that should be considered to further strengthen the conclusions of the work.” The re-organized Figures and additional experiments requested by the reviewer are addressed below.

Major points

- 1. The authors postulate that CD39 inhibition enhances NK cell activity leading to increased recruitment of cDC1s. However, some key experiments required to validate this mechanism are absent. For example confirming that CD39 inhibition fails to enhance cDC1 numbers in the absence of NK cells. NK deficient mice or NK depleting antibodies could be used to confirm this. Do NK cells enhance their expression of XCL1 in the context of CD39 inhibition?**

We thank the reviewer for pointing out that “Some key experiments required to validate this mechanism are absent”. We have used NK cells depletion antibody to conduct the animal experiments. As shown in **SFigure 8A-D** and **Figure 5A-D**, CD39i treatment failed to enhance cDC1 numbers and the effects of CD39i on inhibiting tumor growth and improving the overall survival rate of mice were significantly reversed in the absence of NK cells, indicating that NK cells are the key for CD39i to exert an anti-tumor effect. Additionally, we found that CD39i treatment increased serum XCL1 levels in normal NK cells mice, but not in NK cell absent mice (**SFigure 8E**).

- 2. Figure 4- The NK1.1 staining is not convincing and thus undermines the message that CD39i enhances NK cell infiltrate. The authors should utilise alternative methods of identifying NK cells such as Nkp46 to try and improve this. It may also be interest to concurrently perform a basic analysis of NK cell phenotype too e.g. CD11b, CD27, Granzyme B.**

The reviewer’s point is well taken. We have used Nkp46 to identify NK cells and performed a basic analysis of NK cell phenotype by flow cytometry. As shown in **Figure 4E, F** and **SFigure 6A-G**, CD39i treatment significantly increased the proportion of tumor infiltrated NK cells,

CD11b⁺ NK cells, and CD27⁺ NK cells, but not Granzyme B⁺ NK cells, indicating that the function of NK cells was enhanced.

3. Does CD39 inhibition enhance cDC1 function? Some experiments to show improved T cell priming would be useful to validate this mechanism.

The reviewer requested “Some experiments to show improved T cell priming would be useful to validate this mechanism”. We agree with the reviewer and according to their advice. We have conducted flow cytometry to investigate the effect of CD39i on the proliferation and killing function of intratumoral infiltrating T cells. As shown in **SFigure 7A-H**, CD39i treatment significantly increased the proportion of tumor infiltrated Ki67⁺ T cells, Granzyme B⁺ T cells, Prf1⁺ T cells, but not IFN- γ ⁺ T cells, suggesting that CD39i treatment promoted T cell proliferation and enhanced T cells function by encouraging T cells to secrete more perforin and granzyme B.

4. What effect doses cisplatin have on the immune effects mediated by CD39 inhibition? This should be explored through flow cytometry or RNA analyses. Is there interplay with the cDC1 population for example? Without this detail the rationale to combine these treatment types is not clear.

The reviewers expressed their concern “What effect doses cisplatin have on the immune effects mediated by CD39 inhibition?” and asked “Is there interplay with the cDC1 population for example?” We agree with the reviewer their advice. We have conducted flow cytometry to explore the effect doses cisplatin on the immune effects mediated by CD39i. We found that cisplatin at doses ≥ 3 mg/Kg had a synergistic effect with CD39i on increasing the proportion of cDC1 in tumors, but there was no difference in the synergistic effect of cisplatin at 3mg/Kg or 4mg/Kg (**Figure 7G** and **SFigure 11B**). In addition, we also found that at a cisplatin dose of 4mg/kg, the bodyweight of the mice decreased sharply and they were prone to death. Therefore, the dose of cisplatin selected in this study was 3mg/Kg.

5. The authors have an opportunity to investigate why anti-PD-L1 does not synergise with CD39i in this model. PD-1 blockade is thought to enhance the activation/ expansion of precursor exhausted T cells. This cell population is not annotated in the single cell dataset but can the authors detect changes in this population by scRNA-Seq or FACS following CD39i?

The reviewer’s point is well taken. We have performed flow cytometry to investigate why anti-PD-L1 did not work synergistically with CD39i in this model. As shown in **SFigure 10B, C**, we found that CD39i treatment significantly reduced the number of precursor exhausted T cells, while PD-1 blockade is thought to enhance the activation/expansion of precursor exhausted T cells³³, indicating that both CD39i and aPD-L1 treatments may exert antitumor effects by activating the same cell population.

33. Kallies A, Zehn D, Utzschneider DT. Precursor exhausted T cells: key to successful immunotherapy? Nature reviews Immunology 20, 128-136 (2020).

Minor comments

1. **The conclusions are based entirely upon the CD39 inhibitor POM-1. Whilst POM-1 has reasonable selectivity for CD39, other enzymes are known to be inhibited by this drug and so ideally results would be confirmed using a genetic-based approach. In the absence of this data, I recommend authors simply discuss this potential caveat.**

We thank the reviewer for pointing out that “Whilst POM-1 has reasonable selectivity for CD39, other enzymes are known to be inhibited by this drug”. According to your suggestion, we have stated the potential caveat of POM-1 in the Discussion section as “POM-1 was an inhibitor of ENTPDase, showed the most significant inhibition of CD39/ENTPDase¹³⁵. In this study, POM-1 was used to inhibit CD39, but previous studies showed that POM-1 also inhibited ENTPDase2 and ENTPDase3^{16,35}, which is the shortcoming of this study.”

16. Wall MJ, Wigmore G, Lopatár J, Frenguelli BG, Dale N. The novel NTPDase inhibitor sodium polyoxotungstate (POM-1) inhibits ATP breakdown but also blocks central synaptic transmission, an action independent of NTPDase inhibition. Neuropharmacology 55, 1251-1258 (2008).

35. Müller CE, Iqbal J, Baqi Y, Zimmermann H, Röllich A, Stephan H. Polyoxometalates--a new class of potent ecto-nucleoside triphosphate diphosphohydrolase (NTPDase) inhibitors. Bioorganic & medicinal chemistry letters 16, 5943-5947 (2006).

2. **It was not apparent how frequently flow cytometry analyses were performed e.g. Figures 2D-G as I could not see this information in the Figure legend. As it stands it could be interpreted that this experiment was performed only once, which would be unacceptable.**

We agree with the reviewer and take their advice. We have re-organized the Figures and added a description of flow cytometry analysis frequency in the Methods section as “All the flow cytometry analyses were repeated at least 3 times with 5 samples in each group”.

3. **Figure 3J shows the clustering associated with the distinct myeloid sub populations. But this graph is not annotated to indicate genes characteristic of each cluster and so adds little value to the manuscript in its current form.**

We thank the reviewer for pointing out this issue. **Figure 3J** shows the heterogeneity of different subpopulations of myeloid cells, and we have provided a new picture containing the gene annotations in **Figure 3J** according to your advice.

4. **Line 175 the authors conclude that CD39i promotes expansion of CTL cells but proliferation is not directly assessed and so this data could reflect increased survival. The language here should be modified accordingly.**

By taking the reviewer’s suggestion, we have conducted flow cytometry to investigate the effect of CD39i on the proliferation of intratumoral infiltrating T cells. As shown in **SFigure 7A**,

CD39i treatment significantly increased the proportion of tumor infiltrated Ki67+ T cells, indicating that CD39i treatment promotes T cell proliferation.

5. **Figure 4A- The cell populations on the scRNA-Seq plot do not appear to match the labels below?**

Thanks for raising this important question, and we have modified the **Figure 4A**.

6. **Line 186- The authors suggest Xcl1 release from NK cells may activate cDC1...one assumes this should state “recruit cDC1?”**

The reviewer’s point is well taken. We have modified the language here as “CD39i treatment promoted the release of Xcl1 from NK cells to recruit cDC1”.

Reviewer #4:

We thank the reviewer for the thoughtful comments and the re-organized figure and additional experiments requested by the reviewer are addressed below.

1. **Previous studies have shown CD39 is increased in human bladder cancer including expression in immune cells and scRNA-seq data (PMID: 34381563, 33790578).**

The reviewers expressed their concern “Previous studies have shown CD39 is increased in human bladder cancer including expression in immune cells and scRNA-seq data”. We agree with the reviewer's comment and have carefully read these two articles. Previous studies (PMID: 34381563) confirmed that the high expression of CD39 is involved in bladder tumorigenesis and is correlated with the early stage of bladder cancer. And PMID: 33790578 revealed that the high expression of CD39 tended to result in a worse survival rate by public bioinformatics databases analyses. Through flow cytometry analysis and in vitro cell co-culture, the authors found that the elevated expression of CD39 in CD4+/CD8+ T cells was significantly associated with the pathological T stage, and the CD8+CD39+ T cells have a stronger tumor-killing effect.

In this study, through single cell RNA sequencing analysis of bladder cancer tissue samples, we found that tumor infiltrated CD8+ T cells and NK cells showed a significantly higher level of CD39, along with increased LAG3, another marker of T cell exhaustion (**Figure 1F-G and SFigure 1A**). In the TCGA-BLCA dataset, the CD39 expression level was positively correlated with T cell exhaustion signature (**Figure 1H**). Furthermore, by using a tissue array containing BC and para-cancer tissues, we found that the expression level of CD39 was positively correlated with the proportion of exhausted CD8+ T cells, but was not related to the abundance of CD8+ T cells (**Figure 1K-M & SFigure 3B**), indicating that CD39 is related to T cell dysfunction degree, while not associated with immune cells infiltration level of BC. Based on these findings, we hypothesized that CD39 might play an important role in the regulation of the tumor microenvironment, then a series of experiments were conducted to explore the function and underlying mechanism of CD39 blocking on immune microenvironment of bladder cancer.

Our research perspective is different from the previous two studies, but it does not conflict with each other. Our study and previous studies improved the mechanism and function of CD39 in tumor immune regulation. We have added discussion and analysis of these in the Discussion section. We appreciate the comments and suggestions of the reviewer, which have further strengthened the manuscript.

2. Much of the presented data is not convincing either due to lack of important raw data or experimental controls. All this information could all be included as supplemental information.

By taking the reviewer's suggestion, all the raw data was provided in a single Excel file with data for each figure/table in a separate sheet. Single cell RNA sequencing data was deposited in a publicly accessible database (Gene Expression Omnibus), and accession codes and associated hyperlinks have been provided in the "Data Availability" section as "The single-cell RNA sequencing data of 8 bladder cancer and 3 en para-cancer tissues of bladder cancer patients used in this paper is available in GSA-Human under the accession code HRA000212 and in SRA datasets under BioProject PRJNA662018. TCGA-BLCA datasets were downloaded from UCSC XENA (<http://xena.ucsc.edu/>). In addition, the IMvigor210 dataset²⁰ was exported from the R package IMvigor210CoreBiologies. All the single-cell RNA sequencing data of mice in different treatment groups have been deposited in the Gene Expression Omnibus (GEO) database (<https://www.ncbi.nlm.nih.gov/geo/>) under accession number GSE189127 by password: afahcusgdnixhef. All other relevant raw data and computer code are available in the article, supplementary information, or from corresponding author on reasonable request."

20. Mariathasan S, et al. TGF β attenuates tumour response to PD-L1 blockade by contributing to exclusion of T cells. *Nature* 554, 544-548 (2018).

3. Animal studies use cells implanted S.Q. rather than into the host bladder making extrapolation to human disease difficult. Well established methods for implanting cells to the bladder wall exist.

The reviewer's point is well taken. We have constructed a mouse orthotopic bladder cancer model and explored the therapeutic effect of CD39i. As shown in **Figure 2D-G**, CD39i treatment could significantly reduce the maximum cross-sectional area of the tumor and the bladder weight of mice, proving that treatment with CD39i could significantly inhibit the growth of bladder tumor.

4. **The entire paper was difficult to read including extensive grammar, syntax, and other writing issues. Even the figures are loaded with spelling mistakes (e.g., Fig 2B, vertical axis, “Volumn”)**

We thank the reviewer for raising this important issue. We have re-organized the Figures and have carefully corrected the grammar, syntax, and other writing issues according to your advice.

Specific Issues:

Figure 1

A, how many replicates were completed for these analyses? Please show independent TSNE replicate plots

C, the IHC images are too small to distinguish the cited lineage specific expression of CD39. Pls include low and high magnifications. Please include a low mag scan the CD39 stained TMA in supplemental as well as independent cores each from normal and cancer. At least 5 cores from normal versus tumor.

D, how was this scoring completed? What do values of 5, 10 and 15 represent?

G, this data is not clear as there are a different number of events shown for normal and tumor. Cell numbers should be the same for both TSNE plots.

K, IF staining is not clear especially LAG3 expression. Pls include panels of single channel marker expression, low-high mag views of each stain, and more fields from tumors.

A, By re-analyzing our previously published single cell RNA sequencing data¹⁴, we clustered single cells from 8 tumor samples and 3 para tumor samples of bladder cancer patients into 10 major clusters (**Figure 1A**), the independent TSNE replicate plots were shown in **SFigure 1C**.

C, We agree with the reviewer and take their advice. We have provided IHC images at low and high magnification in the new version of **Figure 1C**. And a low mag scan of the CD39 stained TMA, as well as the independent cores each from normal and cancer (5 cores from normal versus tumor) were shown in **SFigure 2B, C**.

D, The numbers 5, 10, and 15 represent the IRS scores for CD39, and we have modified the Figure accordingly. The scoring method for CD39 was described in detail in the Methods section as “The expression level of CD39 staining was assessed using Remmele and Stegner’s semi-quantitative immunoreactive score (IRS) scale^{67, 68, 69}. Briefly, the CD39 signal was mainly detected in the plasma membrane and extracellular. The staining intensity for CD39 was scored on a scale of 0 ~ 3 (0, negative; 1, weak; 2, moderate; and 3, strong). The CD39 positive cells proportion also was scored as follows: 0, no staining; 1, 1–25%; 2, 26–50%; 3, 51– 75%; and 4, 76–100% stained cells. And then multiplying the staining intensity score and positive cells proportion score to obtain a final score.”

G, We agree with the reviewer’s comments. We contacted the technical analysts of the single cell RNA sequencing company, and they pointed out: these single cell RNA sequencing data are derived from 8 tumor samples and 3 para tumor samples from BC patients, and the cell number of each sample is quite different, so it is impossible to artificially control the cell number to be the same. At present, it is not possible to achieve “Cell numbers were same for both TSNE plots”.

K, The reviewer's point is well taken. We have included panels of single channel marker expression and low-high mag views of each stain with more fields from tumors in **SFigure 3B**.

14. Chen Z, et al. Single-cell RNA sequencing highlights the role of inflammatory cancer-associated fibroblasts in bladder urothelial carcinoma. *Nature communications* 11, 5077 (2020).

67. Remmele W, Stegner HE. [Recommendation for uniform definition of an immunoreactive score (IRS) for immunohistochemical estrogen receptor detection (ER-ICA) in breast cancer tissue]. *Der Pathologe* 8, 138-140 (1987).

68. Tang YH, et al. The long noncoding RNA AK002107 negatively modulates miR-140-5p and targets TGFBR1 to induce epithelial-mesenchymal transition in hepatocellular carcinoma. *Molecular oncology* 13, 1296-1310 (2019).

69. Chen P, Yao Y, Yang N, Gong L, Kong Y, Wu A. Circular RNA circCTNNA1 promotes colorectal cancer progression by sponging miR-149-5p and regulating FOXM1 expression. *Cell death & disease* 11, 557 (2020).

Figure 2

A,H the cartoons are not helpful and should be removed (lack units, incomplete, redundant)

B, please show gross images PBS and POM-1 that match the indicated error bars

C, how is survival defined

D, please include FACS plots for at least 5 ctrl and 5 POM1 tumors with equal events analyzed

I, clusters are not labelled

J, include bar graphs for all ctrl and POM-1 replicates cited in E-G

A, H, By taking the reviewer's suggestion, we have removed the cartoons and have re-organized the Figure accordingly.

B, The reviewer pointed out that our subcutaneous tumor size exceeded ethical limits. Since we did not find the tumor to be as large as imagined during the animal experiments, we carefully examined the raw data and found that we made a mistake in calculating the tumor volume without multiplying by 0.5. This also leads to the deviation of the experimental data. For this reason, we repeated the animal experiments. The mice with bodyweight loss exceeded 20%, becoming moribund, the subcutaneous tumor volumes reached 2000 mm³ or becoming necrotic were considered dead^{56, 57, 58, 59}. The maximal tumour size/burden permitted and the ethical statements are described in the Materials and Methods section as "All animal procedures have been approved by the Animal Care and Use Committee of Tongji Medical College of Huazhong University of Science and Technology (IACUC Number: 2580). The maximal permitted tumour size/burden was 2000 mm³, and when the tumor volume reached 2000 mm³, the mice were euthanized as the humanitarian endpoint of the study." The gross images of subcutaneous tumors were shown in **Figure 2B**.

56. Xu YL, Ding CL, Qian CL, Qi ZT, Wang W. Retinoid acid induced 16 deficiency aggravates colitis and colitis-associated tumorigenesis in mice. *Cell death & disease* 10, 958 (2019).

57. Chongsathidkiet P, et al. Sequestration of T cells in bone marrow in the setting of glioblastoma and other intracranial tumors. *Nature medicine* 24, 1459-1468 (2018).

58. Xiao H, et al. M2-Like Tumor-Associated Macrophage-Targeted Codelivery of STAT6 Inhibitor and IKK β siRNA Induces M2-to-M1 Repolarization for Cancer Immunotherapy with Low Immune Side Effects. ACS central science 6, 1208-1222 (2020).

59. Wang X, et al. Polycarbonate-based ultra-pH sensitive nanoparticles improve therapeutic window. Nature communications 11, 5828 (2020).

D, According to the request of the reviewer, FACS plots for at least 5 ctrl and 5 POM1 tumors with equal events analyzed are shown in **SFigure 4A-C**.

I, The **Figure 2M-N** share the same label.

J, The bar graphs for all ctrl and POM-1 replicates cited in **Figure 1H-L** are shown in **SFigure 4A-C**.

Figure 4

A-B, It is unclear how this shows that T cells are cycling more. Please include individual tumors plots to support this claim

A-B, Suggest using an alternative assay (flow based) to support the claim that CD39 inhibition alters T cell activity.

E, FACS plots show different numbers of events. Pls keep standardize and include at least 5 replicate plots in the supplemental.

H-I, this is completely confused and poorly explained.

A, B, Subcutaneous tumors of 3 mice in NC and POM-1 treatment groups were dissociated to obtain single cell suspensions of immune cells, which were then mixed 1:1:1 for performing single cell RNA sequencing (10X Genomics). In order to further investigate CD39i-mediated changes in T cells, the lymphocytes were clustered into 10 clusters, including CD8-G2M, CD8-S, CD8-Slfn5, CD8-TRM, CD8-Ifitm1, NK, Th, CD4-Pdcd1, Th17, and Treg (**Figure 4A**). We found that the proportion of cycling CD8+ T cells (CD8-G2M and CD8-S) in tumor of POM-1 treated mice was significantly increased compared with NC group (**Figure 4B, D**). Using the same approach, we also found in another single cell RNA sequencing experiment that POM-1 treatment increased the cycling CD8+ T cells within subcutaneous tumors (**Figure 6F, G**).

A, B, We have conducted flow cytometry to investigate the effect of CD39i on the proliferation and killing function of intratumoral infiltrating T cells. As shown in **SFigure 7A-H**, CD39i treatment significantly increased the proportion of tumor infiltrated Ki67+ T cells, Granzyme B+ T cells, Prfl+ T cells, but not IFN- γ + T cells, suggesting that CD39i treatment promoted T cell proliferation and enhanced T cells function by encouraging T cells to secrete more perforin and granzyme B.

E, The reviewer's point is well taken. FACS plots of 5 ctrl and 5 POM1 tumors with equal events analyzed are shown in **SFigure 6A**.

H-I, We agree with the reviewer and take their advice. **Figure 4H-I** was further explained in Results of Manuscript as follows. "In order to investigate whether CD39i influences the communication network between NK cells, cDC1 and T cells, we generated a cell-cell communication network in the subcutaneous tumor with CellPhoneDB 2^{28, 29}, a python-based database of cell receptors, ligands, and

their interactions that can be used to study cell-to-cell interactions at the molecular level. We found cDC1 and mDC1 expressed Il15, Il12b, Cd80 and Cd86, which might enhance the antitumor activity of cycling CD8+ T cells by interacting with Il15ra, Il12rb1/2, and Cd28 (**Figure 4H**). While previous studies have shown that Il12b-secreting mDC is mainly derived from cDC1^{21, 30}. NK cells communicated with cDC1 by expressing Xcl1, which leads to the chemotaxis of cDC1 into the tumor, and CD39i can enhance this effect by increasing NK cells in the tumor. Additionally, we also observed that cDC1 and mDC1 could negatively regulate the function of cycling CD8+ T cells by expressing CD274 (Pd-1), Lgals9, and Tnfsf9 (**Figure 4H**). Collectively, we hypothesized that CD39i treatment promoted the release of Xcl1 from NK cells to recruit cDC1, which activated cycling CD8+ T cells through expressing Il15, Il12b, Cd80 and Cd86 when it matured, thus achieving the anti-tumor effect (**Figure 4I**). Moreover, this process is negatively regulated by immune checkpoints.”

21. Maier B, et al. A conserved dendritic-cell regulatory program limits antitumour immunity. *Nature* 580, 257-262 (2020).
28. Vento-Tormo R, et al. Single-cell reconstruction of the early maternal–fetal interface in humans. *Nature* 563, 347-353 (2018).
29. Efremova M, Vento-Tormo M, Teichmann SA, Vento-Tormo R. CellPhoneDB: inferring cell–cell communication from combined expression of multi-subunit ligand–receptor complexes. *Nature Protocols* 15, 1484-1506 (2020).
30. Cheng S, et al. A pan-cancer single-cell transcriptional atlas of tumor infiltrating myeloid cells. *Cell* 184, 792-809.e723 (2021).

Figure 5.

A, cartoon is not clear and redundant. Suggest removing

B, tumor growth curves are not discernable as presented. Either fix the vertical scale or format. Where are the error bars?

C, please show the absence of Batf3 by IHV to confirm that the expression difference between WT and KO mice using implanted tumors. Show low and high mag images of the MB tumors implanted to WT and KO mice.

B-C, the IACUC limit for individual SQ tumors in mice is 2000 mm³, thus is unclear how the authors have produced tumors (individual data points) of tumors exceeding 4000mm³ and up to 6000mm³. Please show gross images of these SQ tumors and how these related to the error bars shown.

A, At the request of the reviewer, we have removed the cartoons.

B, These curves represent the individual tumor growth curve of WT, WT+CD39i, Batf3^{-/-}, Batf3^{-/-}+CD39i group, and we have re-organized the **SFigure 9A** according to the suggestions of reviewer. The gross tumor growth curve with error bars were shown in **Figure 5E**.

C, To confirm the expression difference of BATF3 between WT and Batf3 KO mice, we have performed IHC using implanted tumors. As shown in **SFigure 8F**, BATF3 positive cells were present in implanted tumors of WT mice, but not Batf3 KO mice.

B-C, The reviewer pointed out that our subcutaneous tumor size exceeded ethical limits. Since we did not find the tumor to be as large as imagined during the animal experiments, we carefully examined the raw data and found that we made a mistake in calculating the tumor volume without multiplying by 0.5. This also leads to the deviation of the experimental data. For this reason, we repeated the animal experiments. The gross images of subcutaneous tumors of mice were shown in **Figure 5F**.

Sincerely,
Ke Chen, MD, PhD,
Department of Urology,
Wuhan Union Hospital, Tongji Medical College,
Huazhong University of Science and Technology

Reviewers' Comments:

Reviewer #1:

Remarks to the Author:

In the revised manuscript the authors have addressed the concerns I raised upon the initial submission. I have no further reservations about the manuscript. However, I would recommend a thorough grammar check before publication.

Reviewer #3:

Remarks to the Author:

The authors have performed several new experiments/ analyses that add significant value to the manuscript. However, in some cases the conclusions made are not supported by the data, at least as they are currently presented. There are therefore additional modifications/ amendments that must be considered.

1) Link between NK cells and cDC1-

The new data indicating that CD39i is ineffective in NK cell or cDC1 deficient mice support a link between these cell types. However some questions remain-

a) the gating strategy used to define cDC1 is unclear, this should be shown. The authors appear to have defined cDC1 based upon lack of expression of CD11b and CD172. It is not clear what markers the authors used to exclude other cell types and whether the expression of CD103 and XCR1 was determined by FACS- these really should be included for analyzing this cell type

b) The authors conclude in the text that CD39i enhances NK cell activation and expression of XCL1. In fact no data is shown for NK cell expression of XCL1 and the expression of activation markers on NK cells appears to be comparable between treated and non treated mice. Can the authors investigate XCL1 expression (and other chemokines as comparators) by NK cells in control vs CD39i treated mice from their scRNA-seq data? Differences in the percentage of Granzyme B+NKp46+ cells appear to be driven entirely by increased NK cell numbers and not increased Granzyme B expression/activation on a per cell basis. These conclusions need to be modified. The CD11b/ CD27 analysis is also curious. Conventionally these markers are analysed together to subset NK cells into mature/immature cells. It appears that virtually all NK cells are CD11b+CD27+ based on the data presented which is also odd.

c) Why do the authors investigate XCL1 expression in the serum and not the tumour, which would be more relevant for cDC1 trafficking to the tumour?

d) The authors perform scRNA-seq on CD39i treated tumours from WT and Batf KO mice but do not really conclude on the differences observed. It looks as if, analysing by eye, that CD39i results in increased NK cell numbers in both mice strains but the effects on CD8+ T cells are ablated in cDC1 deficient mice. This would be an important conclusion to highlight as it adds strength to the proposed model whereby CD39i indirectly activates T cells through the cDC1/ NK axis.

2) Link between cisplatin and cDC1s.

The authors conclude that there is a synergistic interaction between cisplatin and CD39i based on the proportion of cDC1. However the control of cisplatin alone is not compared to combination treated mice, and so this conclusion is not valid based on the data presented.

3) From the scRNAseq data one of the clearest effects on CD8+ T cells appears to be increased expression of a gene cluster annotated as "Exhausted related genes" (Figure 4G) but the authors do not really discuss this observation.

4) The authors state in the text that “cDC1 and mDC1 could negatively regulate the function of cycling CD8+ T cells” due to their expression of PD-L1....This should probably say “potentially negatively regulate” given that the authors have not experimentally determined this.

5) Line 229 refers to a xenograft model but this is not technically a xenograft model as this term refers to transplantation of a tumour derived from one species into another.

6) In their rebuttal letter the authors highlight that because both anti-PD-1 and CD39i reduce the proportion of precursor exhausted cells it may undermine their potential to act synergistically. This is intriguing but this was not clearly highlighted in the relevant section of the manuscript. This discussion should be added.

Reviewer #4:

Remarks to the Author:

Through the inclusion of additional controls, raw data new in vivo studies, the manuscript has improved significantly. The authors have also provided alternative approaches to relate CD39 inhibition with altered T cell activity (proliferation, exhaustion) and tumor progression. The following major concern must be reconciled.

Figure 1A, Figure S1C. The TSNE plots shown reveal dramatic variability in epithelial-immune cells present between tumors including some with very low cancer cells present (T1, T2, T5, T7). Such tumors appear to have, on a relative scale, less epithelia than the 3 normal bladders. Moreover, the number of infiltrating T-NK cells (green) shown is the same or less than the tumors. This data is concerning given that the presence of tumor cells drives the T-NK cell exhaustion and immune cell infiltration drives response to therapies including CD39i.

Minor

Figure. 1G. The TSNE plots shown do not provide quantitative data. Consider to gate on NK-T cells populations and show the numbers of LAG3, CD39 positive cells.

Re: NCOMMS-21-40590A

Each of the specific questions raised by the reviewers is described in point form below.

Reviewer #1:

We thank the reviewer for the recognition of our manuscript. We have checked the grammar again according to your suggestion.

Reviewer #3:

We thank the reviewer for the thoughtful comments that “The authors have performed several new experiments/ analyses that add significant value to the manuscript.” And agree with the reviewer that “There are therefore additional modifications/ amendments that must be considered.” The issues raised by the reviewer have been addressed below.

1) Link between NK cells and cDC1-

The new data indicating that CD39i is ineffective in NK cell or cDC1 deficient mice support a link between these cell types. However some questions remain-

a) the gating strategy used to define cDC1 is unclear, this should be shown. The authors appear to have defined cDC1 based upon lack of expression of CD11b and CD172. It is not clear what markers the authors used to exclude other cell types and whether the expression of CD103 and XCR1 was determined by FACS- these really should be included for analyzing this cell type

The reviewers expressed their concern “It is not clear what markers the authors used to exclude other cell types and whether the expression of CD103 and XCR1 was determined by FACS-” and asked “the gating strategy used to define cDC1 is unclear, this should be shown.” We agree with the reviewer’s advice and have added CD103 and XCR1 to identify NK cells and performed flow cytometry. As shown in **Figure 5D** and **SFigure 8E**, CD39i treatment increased the proportion (%) of tumor infiltrated cDC1 in control mice, but not in NK cell-depleted mice. We found CD39i treatment increased the proportion (%) of XCR1+ cDC1 in tumors of WT mice (**SFigure 8F, G**). In addition, the same gating strategy used to define cDC1 was also used to explore the synergistic effect of CD39i and cisplatin by flow cytometry. We found that CD39i (but not cisplatin alone) treatment significantly increased the proportion of tumor infiltrated cDC1 cells. And cisplatin at 3mg/Kg had a synergistic effect with CD39i on increasing the proportion of cDC1 in tumors (**Figure 7G** and **SFigure 11D**). The gating strategy (CD45+CD11c+MHC II+CD103+CD11b-XCR1+) used to define cDC1 is shown in **SFigure 8D**.

b) The authors conclude in the text that CD39i enhances NK cell activation and expression of XCL1. In fact no data is shown for NK cell expression of XCL1 and

the expression of activation markers on NK cells appears to be comparable between treated and non treated mice. Can the authors investigate XCL1 expression (and other chemokines as comparators) by NK cells in control vs CD39i treated mice from their scRNA-seq data? Differences in the percentage of Granzyme B+NKp46+ cells appear to be driven entirely by increased NK cell numbers and not increased Granzyme B expression/activation on a per cell basis. These conclusions need to be modified. The CD11b/ CD27 analysis is also curious. Conventionally these markers are analysed together to subset NK cells into mature/immature cells. It appears that virtually all NK cells are CD11b+CD27+ based on the data presented which is also odd.

The reviewer requested “Can the authors investigate XCL1 expression by NK cells in control vs CD39i treated mice from their scRNA-seq data?” And expressed their concern “It appears that virtually all NK cells are CD11b+CD27+ based on the data presented which is also odd.” We agree with the reviewer their advice.

Firstly, we investigated the XCL1 expression by NK cells in control vs CD39i treated mice from the scRNA-seq data. As shown in **SFigure 6F**, CD39i treatment did not change the expression level of XCL1 on a per cell basis, suggesting that NK cells may achieve anti-tumor effect by increasing cell number rather than enhancing XCL1 expression in individual cells.

Secondly, we have modified these conclusions as “We found no difference in the proportion of Granzyme B+ NK cells between control and CD39i treated group (**SFigure 6D, E**), suggesting the differences in the percentage of Granzyme B+ NK cells appear to be driven entirely by increased NK cell numbers and not increased Granzyme B expression/activation on a per cell basis.”

Thirdly, we have conducted flow cytometry to investigate the proportion of mature/immature NK cells. As shown in **SFigure 6B, C**, CD39i treatment significantly increased the proportion of CD27+CD11b+ and CD27-CD11b+ NK cells, but decreased the proportion of CD27-CD11b- and CD27+CD11b- NK cells.

c) Why do the authors investigate XCL1 expression in the serum and not the tumour, which would be more relevant for cDC1 trafficking to the tumour?

We agree with the reviewer and take their advice. We have investigated XCL1 protein expression in the tumor by ELISA, the results showed that CD39i treatment increased tumor XCL1 protein levels in control mice, but not in NK cells absent mice (**SFigure 8H**).

d) The authors perform scRNA-seq on CD39i treated tumours from WT and Batf KO mice but do not really conclude on the differences observed. It looks as if, analysing by eye, that CD39i results in increased NK cell numbers in both mice

strains but the effects on CD8+ T cells are ablated in cDC1 deficient mice. This would be an important conclusion to highlight as it adds strength to the proposed model whereby CD39i indirectly activates T cells through the cDC1/ NK axis.

Thanks to the reviewer for putting forward these crucial revision suggestions, which are very helpful for us to further improve the manuscript. According to the reviewer's suggestion, we added this important conclusion "We observed that CD39i treatment significantly increased the proportion of proliferating CD8+ T cells and 5 subtypes of NK cells, along with a slight decrease in Treg abundance in WT mice, the same as that observed in the first batch (**Figure 6G**). CD39i treatment increased NK cell numbers in *Batf3*^{-/-} cDC1-deficient mice, but the effects on CD8+ T cells are ablated (**Figure 6G**), indicating that CD39i indirectly activates T cells through the NK/cDC1 axis" in the Results section of the manuscript.

2) Link between cisplatin and cDC1s.

The authors conclude that there is a synergistic interaction between cisplatin and CD39i based on the proportion of cDC1. However the control of cisplatin alone is not compared to combination treated mice, and so this conclusion is not valid based on the data presented.

Thank the reviewers for pointing out this issue. We have performed flow cytometry to verify the synergistic effect of cisplatin (3mg/Kg) and CD39i. We found that CD39i (but not cisplatin alone) treatment significantly increased the proportion of tumor infiltrated cDC1 cells. And cisplatin at 3mg/Kg had a synergistic effect with CD39i on increasing the proportion of cDC1 in tumors ((**Figure 7G** and **SFigure 11D**). The gating strategy (CD45+CD11c+MHC II+CD103+CD11b-) used to define cDC1 is shown in **SFigure 8D**.

3) From the scRNAseq data one of the clearest effects on CD8+ T cells appears to be increased expression of a gene cluster annotated as "Exhausted related genes" (Figure 4G) but the authors do not really discuss this observation.

Thanks to the reviewers for pointing out this obvious flaw. By taking the reviewer's suggestion, we discussed this observation in the Results section as "Interestingly, we also observed that co-stimulatory and co-inhibitory checkpoints are almost absent in CD8+ T cells except for the cycling cells (**Figure 4G**), indicating that CD39i treatment did not completely reverse the tumor immunosuppressive microenvironment, that is, while CD39i enhances the cytotoxicity of cycling CD8+ cells, the anticancer function of these cells might also be inhibited by other negative regulatory mechanisms, such as PD1/PD-L1 pathway." And in Discussion section "we found that although CD39i greatly promoted CTL amplification and cytotoxicity, it was also accompanied by a remarkable increase in co-inhibitory receptors, indicating that the anticancer function

of CD39i may be limited by other negative regulatory mechanisms.”

4) The authors state in the text that “cDC1 and mDC1 could negatively regulate the function of cycling CD8+ T cells” due to their expression of PD-L1....This should probably say “potentially negatively regulate” given that the authors have not experimentally determined this.

The reviewer’s point is well taken. We have modified the language here as “we also observed that cDC1 and mDC1 could potentially negatively regulate the function of cycling CD8+ T cells by expressing CD274 (PD-L1), LGALS9, and TNFSF9.”

5) Line 229 refers to a xenograft model but this is not technically a xenograft model as this term refers to transplantation of a tumour derived from one species into another.

Thank the reviewer for pointing out this issue. According to your suggestion, we have modified the language here as “As described before, tumor models were constructed by subcutaneous injection of MB49 cells into *Batf3*^{-/-} or *Batf3*^{+/+} (WT) C57BL/6J mice.”

6) In their rebuttal letter the authors highlight that because both anti-PD-1 and CD39i reduce the proportion of precursor exhausted cells it may undermine their potential to act synergistically. This is intriguing but this was not clearly highlighted in the relevant section of the manuscript. This discussion should be added.

We agree with the reviewer and take their advice. We have added the discussion about this intriguing finding as “we found that CD39i treatment significantly reduced the number of precursor exhausted T cells (**SFigure 10B, C**), while aPD-1 is thought to enhance the activation/expansion of precursor exhausted T cells³³, which may undermine their potential to act synergistically” in the Results section.

And in the Discussion section as “After flow cytometry analysis, we found that CD39i treatment conspicuously reduced the number of precursor exhausted T cells, a population that was thought to be susceptible to PD-1 blockade-mediated activation/expansion³³. Therefore, we speculate that both CD39i and aPD-L1 treatments may exert anti-tumor effects by affecting the same cell population, which may undermine their potential to act synergistically.”

33. Kallies A, Zehn D, Utzschneider DT. Precursor exhausted T cells: key to successful immunotherapy? *Nature reviews Immunology* 20, 128-136 (2020).

Reviewer #4:

We thank the reviewer for the recognition of our revised manuscript “Through the inclusion of additional controls, raw data new in vivo studies, the manuscript has

improved significantly.” And the remaining issues raised by the reviewer are addressed below.

Figure 1A, Figure S1C. The TSNE plots shown reveal dramatic variability in epithelial-immune cells present between tumors including some with very low cancer cells present (T1, T2, T5, T7). Such tumors appear to have, on a relative scale, less epithelia than the 3 normal bladders. Moreover, the number of infiltrating T-NK cells (green) shown is the same or less than the tumors. This data is concerning given that the presence of tumor cells drives the T-NK cell exhaustion and immune cell infiltration drives response to therapies including CD39i.

We thank the reviewer for the thoughtful comments which are very helpful for us to further improve the manuscript. We agree with the reviewer their advice.

The effects of CD39i, as well as other means of immunotherapy, largely depend on the degree of immune cell infiltration, which is in line with current achievements in the field of bladder cancer immunotherapy. For immune desert type tumors, promoting immune infiltration by small molecule drugs can significantly reverse the problem of immunotherapy tolerance. For tumor tissues, the proportion of immune cells in bladder cancer depends on its molecular subtype, which was discussed in our previous paper¹⁴. Tumors of different molecular subtypes could vary in their degree of T-cell infiltration several- or even dozens of folds. Therefore, the large difference in the proportion of epithelial-immune cells among the 8 tumor samples might be a manifestation of the differences in tumor molecular typing. Generally, immune exclusion occurs in luminal-papillary type of bladder cancer with few immune cells and stromal cells (T4, T8), in this part of tumor tissue, which belongs to immune noninvasive type tumor, not immune depletion tumor. Meanwhile, current clinical studies have found that anti-PD-L1 therapy is only suitable for PD-L1+ patients, and for patients with immune exclusive, how to promote the entry of immune cells into the tumor remains a great challenge. Our study found that high CD39 expression was usually accompanied by enrichment of CD8+LAG3+ exhausted T cells, indicating the amount of CD39 expression was positively correlated with T cell exhaustion. Meanwhile, through reanalysis of Imvigor210 RNA-seq dataset²⁰ we found that patients with moderate expression of PD-L1 and high expression of CD39 responded poorly to treatment with anti-PD-L1, which might be the preferred population for CD39i treatment strategies. Therefore, taken together with the results of in vivo experiments, we propose that CD39i treatment might be a better choice for patients with higher CD39 expression, especially in patients who fail to respond to anti-PD-1 or anti-PD-L1 therapy.

We have added this part in the Discussion section of the manuscript.

14. Chen Z, et al. Single-cell RNA sequencing highlights the role of inflammatory cancer-associated fibroblasts in bladder urothelial carcinoma. *Nature communications* 11, 5077 (2020).

20. Mariathasan S, et al. TGF β attenuates tumour response to PD-L1 blockade by contributing

to exclusion of T cells. Nature 554, 544-548 (2018).

Figure. 1G. The TSNE plots shown do not provide quantitative data. Consider to gate on NK-T cells populations and show the numbers of LAG3, CD39 positive cells.

The reviewer expressed their concern “The TSNE plots shown do not provide quantitative data.” And requested “Consider to gate on NK-T cells populations and show the numbers of LAG3, CD39 positive cells.” We agree with the reviewer their advice. We investigated the CD39 and LAG3 expression by lymphocytes in Tumor vs Normal from the scRNA-seq data. As shown in **SFigure 2D**, the proportion of CD39-positive lymphocytes and LAG3-positive lymphocytes derived from tumor tissues were significantly higher than that of normal tissues, indicating the amount of CD39 expression was positively correlated with T cell exhaustion.

Reviewers' Comments:

Reviewer #3:

Remarks to the Author:

Thank you for addressing my questions. Some remaining concerns that should be addressable with reanalysis of existing data are outlined below.

The authors have now included CD103 and XCR1 markers as suggested (thank you) but the data are still analysed unsatisfactorily. In Figure 5D and 7G the analysis presented is (presumably) the percentage of CD45+CD11c+MHCII+ cells that are CD11b-CD103+. The key data would what proportion of total CD45+ cells exhibit this phenotype to exclude potential changes to the CD11c+MHCII+ cells affecting the analysis. Similarly in Supplementary Figure 8G the proportion of CD11b+CD103+ cells expressing XCR1 is indicated whereas it would be more relevant again to report this as a percentage of the total CD45+ population.

Thank you for including new scRNAseq data for xcl1 expression. Based on this data I believe the following sentence should be revised: "Collectively, we hypothesized that CD39i treatment promoted the release of Xcl1 from NK cells to recruit cDC1"

To my mind this infers that XCL1 would be increased on a per cell basis but it is actually just more NK cells expressing similar levels of XCL1.

Reviewer #4:

Remarks to the Author:

Figure 1A. The concern over the dramatic variability in epithelial-immune cells in the 8 scRNA seq data sets remains. The authors present these data as an aggregate TSNE plot (Figure 1) however this plot is markedly different than any single tumor raw data set collected (i.e. Fig 1A is not a representative plot of the raw data in Fig S1C).

The effects of CD39i are dependent upon the degree of immune cell infiltration but based on the data sets shown CD39 there are more NK-T cells in the normal plots.

The authors conjecture that subtyping may be the cause for variability but do not include any simple analysis to support this suggestion including accepted markers (CK8 luminal, CK5 basal etc.). Moreover, CD39 also wasn't included. Please include CD39 and LAG3 TSNE expression plots for each human plot shown in Figure S1C using a standardized (equal) cell count between plots.

Re: NCOMMS-21-40590B

Each of the specific questions raised by the reviewers is described in point form below.

Reviewer #3:

Thank you for addressing my questions. Some remaining concerns that should be addressable with reanalysis of existing data are outlined below.

The authors have now included CD103 and XCR1 markers as suggested (thank you) but the data are still analysed unsatisfactorily. In Figure 5D and 7G the analysis presented is (presumably) the percentage of CD45+CD11c+MHCII+ cells that are CD11b-CD103+. The key data would what proportion of total CD45+ cells exhibit this phenotype to exclude potential changes to the CD11c+MHCII+ cells affecting the analysis. Similarly, in Supplementary Figure 8G the proportion of CD11b+CD103+ cells expressing XCR1 is indicated whereas it would be more relevant again to report this as a percentage of the total CD45+ population.

Thank you for including new scRNAseq data for xcl1 expression. Based on this data I believe the following sentence should be revised: “Collectively, we hypothesized that CD39i treatment promoted the release of Xcl1 from NK cells to recruit cDC1”

To my mind this infers that XCL1 would be increased on a per cell basis but it is actually just more NK cells expressing similar levels of XCL1.

The reviewer’s point is well taken. The percentage of cDC1 (CD45+CD11c+MHCII+CD11b-CD103+) and XCR1+ cDC1 in total CD45+ cells were shown in **SFigure 10E, 10G** and **SFigure 13D**. And the sentence “Collectively, we hypothesized that CD39i treatment promoted the release of Xcl1 from NK cells to recruit cDC1.....” have been revised as “Collectively, our results suggested that CD39i treatment increased NK cell infiltration, and then released more Xcl1 (in total rather than on a per cell basis) to recruit cDC1.....” Thank you again for your valuable advice.

Reviewer #4:

Figure 1A. The concern over the dramatic variability in epithelial-immune cells in the 8 scRNA seq data sets remains. The authors present these data as an aggregate TSNE plot (Figure 1) however this plot is markedly different than any single tumor raw data set collected (i.e. Fig 1A is not a representative plot of the raw data in Fig S1C).

The effects of CD39i are dependent upon the degree of immune cell infiltration but based on the data sets shown CD39 there are more NK-T cells in the normal plots.

The authors conjecture that subtyping may be the cause for variability but do not include any simple analysis to support this suggestion including accepted markers (CK8 luminal, CK5 basal etc.). Moreover, CD39 also wasn’t included. Please include CD39 and LAG3 TSNE expression plots for each human plot shown in Figure S1C using a standardized (equal) cell count between plots.

We agree with the reviewer and take their advice. The single-cell clustering results of the 11 samples were quite different. To better show the cell subclasses, the **Figure 1A** mentioned by the reviewer shows the clustering diagram of all the single cells collected from the 11 samples.

The reviewers expressed their concern “The effects of CD39i are dependent upon the degree of immune cell infiltration but based on the data sets shown CD39 there are more NK-T cells in the normal plots.” We agree with the reviewer. Using GEPIA2, the more immune cells in the paracancerous tissues were also present in the TCGA-BLCA data (**SFigure 2D-E**). We have discussed this phenomenon in previous article¹⁴. Since most of the immune cells in the normal bladder and bladder cancer are located in the stroma, the thin transitional epithelium of the normal bladder during sampling leads to the inclusion of a large amount of interstitial tissue in the paracancerous tissue, thus finally forming the manifestation of more immune cells and stromal cells in the paracancerous tissue. This is consistent with the results of bulk RNA-seq (**SFigure 2D-E**).

According to the reviewer’s suggestion, we normalized the number of cells in each sample, taking 2000 single cells randomly from the data in each sample, then the single cells from 8 BC and 3 paracancer tissues were clustered into 10 major clusters (**SFigure 2B**). The **SFigure 2C** shows the clustering diagram of 22, 000 single cells collected randomly from the 11 samples. We detected the expression of CK5 and CK8 in each sample (**SFigure 3A**), with only one sample exhibiting basal characteristics and the remainder exhibiting partial basal differentiation. Due to the lack of bulk-seq data, we were unable to accurately classify the molecular subtypes. We visualized CD39 and LAG3 as individual samples with equal cell count. The results suggest that significant up-regulation of T cell surface CD39 and LAG3 expression in the tumor, which did not occur in all tumor samples but was observed in most tumor samples (**SFigure 3A & SFigure 4B**). Similarly, although CD39 was expressed in paracancerous stromal and endothelial cells, the levels were lower than that in tumor-derived cells (**SFigure 2F**). This is consistent with our immunohistochemical staining results (**Figure 1C-D & SFigure 3B-C**).

14. Chen Z, et al. Single-cell RNA sequencing highlights the role of inflammatory cancer-associated fibroblasts in bladder urothelial carcinoma. *Nature communications* 11, 5077 (2020).